# Agent-Centric State Discovery for Finite-Memory POMDPs

## Abstract

Discovering an informative, or agent-centric, state representation that encodes only the relevant information while discarding the irrelevant is a key challenge towards scaling reinforcement learning algorithms and efficiently applying them to downstream tasks. Prior works studied this problem in high-dimensional Markovian environments, when the current observation may be a complex object but is sufficient to decode the informative state. In this work, we consider the problem of discovering the agent-centric state in the more challenging high-dimensional non-Markovian setting, when the state can be decoded from a sequence of past observations. We establish that generalized inverse models can be adapted for learning agent-centric state representation for this task. Our results include asymptotic theory as well as negative results for alternative intuitive algorithms, such as encoding with only a forward-running sequence model. We complement these findings with a thorough empirical study on the agent-centric state discovery abilities of the different alternatives we put forward. Particularly notable is our analysis of past actions, where we show that these can be a double-edged sword: making the algorithms more successful when used correctly and causing dramatic failure when used incorrectly.

## 1 Introduction

Reinforcement Learning (RL) and its associated tasks of planning and exploration are dramatically easier in a small Markovian state space than in a high-dimensional, Partially Observed Markov Decision Process (POMDP). For example, controlling a car from a set of coordinates and velocities is much easier than controlling a car from first-person camera images. This gap explains why many of the successes of RL have come in domains where this minimal Markovian state space is given or is simple to learn. At the same time, this gap suggests an opportunity:

*Can we devise an unsupervised learning algorithm for automatically discovering the informative Markovian state space from a rich non-Markovian observation space?*

Having such capability has many merits. It can allow faster adaptation for downstream tasks, it simplifies the debugging of the learned representation, and it enables the use of large corpuses of unsupervised datasets in an efficient manner. Learning to extract effective information in complex control systems can be notoriously difficult in general. Indeed, in recent years, much effort has been devoted to tackling this problem in high-dimensional and Markovian systems in the RL community Li et al. (2006); Nachum et al. (2018); Misra et al. (2020); Zhang et al. (2020); Efroni et al. (2022d); Wang et al. (2022b). However, in many real-world control and decision problems, the immediate observation does not contain the complete relevant information required for optimal behavior, and the environment may be non-Markovian. Hence, in practice, an algorithm designer often faces a double challenge: learning in the presence of both non-Markovian and high-dimensional data. To the best of our knowledge, no prior work has focused on developing techniques for state space discovery in such a setting.

In this work, we take a first step towards a solution for the general problem by considering a special and prevalent class of non-Markovian environments. We consider the class of POMDPs with finite-memory, which we refer to as FM-POMDPs, and design techniques to recover the informative state in a high-dimensional setting. Intuitively, for an FM-POMDP, the state can be decoded from a finite sequence of past observations and is often encountered in control and decision problems

(e.g., to decode the velocity or acceleration of an object, a few previous observations are required). Due to the significance of such a system, many past works have put forward techniques for solving and learning in decision problems with memory, both in practice and theory (Bakker, 2001; Graves et al., 2016; Pritzel et al., 2017; Efroni et al., 2022b; Liu et al., 2022; Zhan et al., 2022)). Yet, none explicitly focused on state discovery.

Provably capturing relevant agent states while discarding distractors in Markovian environments has become a widely studied problem in both reinforcement learning theory (Efroni et al., 2022d;a; Wang et al., 2022c) and experimentation (Islam et al., 2023; Lamb et al., 2022; Wang et al., 2022b). These works have demonstrated the ability to discover an *agent-centric state*, which captures all information that is relevant to the control of the agent while discarding *exogenous noise*, the information that is present in the observations but unrelated to the control of the agent. The multi-step inverse kinematics algorithm, studied in these prior works, consists of predicting the first action that an agent takes between an initial observation and an observation after random $k$ steps into the future; predicting an action that caused a future observation from current state. Inverse kinematics has become a popular approach as it is relatively easy to implement and has theoretical guarantees for filtering exogenous noise. At the same time, direct implementation of this approach fails for the FM-POMDP setting because the current observation may be insufficient to encode the state.

We first show that naive approaches to generalize multi-step inverse kinematics can fail, both theoretically and empirically in the FM-POMDP setting. For instance, if a sequence is encoded using an RNN (or any other directed sequence model) and the hidden states are used to predict actions, we show that there is an "action recorder" problem where the model can learn shortcuts to representing the true state. Under assumptions of past and future decodability, we generalize inverse models to the high-dimensional FM-POMDP setting and establish, empirically and theoretically, that it recovers the latent state. Our results show that our variant of the multi-step inverse model can indeed succeed in the FM-POMDP setting. This considers the information of sequences of observations and actions, from both the past and the future, where we can use a forward-backward sequence model to learn agent-centric representations. Experimentally, we validate precise recovery of the state on acceleration-control, information masking, first-person perspective control, and delayed signal problems. Finally, we also demonstrate the usefulness of the proposed objectives in visual offline RL tasks in presence of exogenous information, where we mask out randomly stacked frames and add random masking of patches to learn representations in a partially observable offline RL setting.

## 2 BACKGROUND AND PRELIMINARIES

**Proposed Setting.** We consider a finite-memory POMDP in which the state is assumed to be decodable from a short history of observations (Efroni et al., 2022b). Further, as in many applications in practice, we assume the observations are in high-dimension. This may give rise to the presence of exogenous noise with non-trivial time correlation. This setting can be modelled as an Exogenous Block MDP (Efroni et al., 2022d; Lamb et al., 2022), a rich observation setting in which the observations consists of an agent-centric and exogenous part of the state. This definition assumes that exogenous noise has no interaction with the agent-centric state, which is stricter than the exogenous MDP definition introduced by Dietterich et al. (2018). Our proposed setting combines these to consider an Agent-Centric FM-POMDP in which we assume that the agent-centric state is decodable from a short history of observations.

**Agent-Centric FM-POMDP.** We consider a finite-memory episodic Partially Observable Markov Decision Process (FM-POMDP), which can be specified by $\mathcal{M} = (\mathcal{S}, \Xi, \mathcal{O}, \mathcal{A}, H, \mathbb{P}, \mathbb{O}, r)$. Here $\mathcal{S}$ is the unobservable agent-centric state space, $\Xi$ is the unobservable exogenous state space (for convenience, $\mathcal{Z} = \mathcal{S} \times \Xi$), $\mathcal{O}$ is the observation space, $\mathcal{A}$ is the action space, and $H$ is the horizon. $\mathbb{P} = \{\mathbb{P}_h\}_{h=1}^H$ is a collection of *unknown* transition probabilities with $\mathbb{P}(z' \mid z, a)$ equal to the probability of transitioning to $z'$ after taking action $a$ in state $z$. $\mathbb{O} = \{\mathbb{O}_h\}_{h=1}^H$ are the *unknown* emissions with $\mathbb{O}_h(o \mid s, \xi)$ equal to probability that the environment emits observation $o$ when in agent-centric state $s$ and exogenous state $\xi$ at the $h^{\text{th}}$ step. The *block assumption* holds if the support of the emission distributions of any two states are disjoint, $\text{supp}(q(\cdot \mid z_1)) \cap \text{supp}(q(\cdot | z_2)) = \emptyset$ when $z_1 \neq z_2$., where $\text{supp}(q(\cdot \mid z)) = \{o \in \mathcal{O} \mid q(o \mid z) > 0\}$ for any state $z$. We assume our action space is finite and our agent-centric state space is also finite.

The agent-centric FM-POMDP is concerned with the structure of the state space $\mathcal{Z}$. More concretely, the state space $\mathcal{Z} = \mathcal{S} \times \Xi$ consists of an agent-centric state $s \in \mathcal{S}$ and $\xi \in \Xi$, such that $z = (s, \xi)$. The state dynamics are assumed to factorize as $\mathbb{P}(s', \xi'|s, \xi, a) = \mathbb{P}(s'|s, a)\mathbb{P}(\xi'|\xi)$. We do not consider the episodic setting, but only assume access to a single trajectory. The agent interacts with the environment, generating an observation and action sequence, $(z_1, o_1, a_1, z_2, o_2, a_2, \cdots)$ where $z_1 \sim \mu(\cdot)$. The latent dynamics follow $z_{t+1} \sim T(z' \mid z_t, a_t)$ and observations are generated from the latent state at the same time step: $o_t \sim q(\cdot \mid z_t)$. The agent does not observe the latent states $(z_1, z_2, \cdots)$, instead it receives only the observations $(o_1, o_2, \cdots)$. We use $\tilde{\mathcal{O}}_m$ to denote the set of augmented observations of length $m$ given by $\tilde{\mathcal{O}}_m = (\mathcal{O} \times \mathcal{A})^m \times \mathcal{O}$. Moreover, we will introduce the notation that $\tilde{o}_t = (o_t, a_{t-1})$, which can be seen as the observation augmented with the previous action. Lastly, the agent chooses actions using a policy which can most generally depend on the entire $t$-step history of observations and previous actions $\pi : \tilde{\mathcal{O}}^t \to \Delta(\mathcal{A})$, so that $a_t \sim \pi(\cdot|\tilde{o}_1, ..., \tilde{o}_{t-1}, \tilde{o}_t)$.

We assume that the agent-centric dynamics are deterministic and that the diameter of the control-endogenous part of the state space is bounded. In other words, there is an optimal policy to reach any state from any other state in a finite number of steps: the length of the shortest path between any $s_1 \in \mathcal{S}$ to any $s_2 \in \mathcal{S}$ is bounded by $D$. These assumptions are required for establishing the theoretical guarantees in Lamb et al. (2022), we built upon in this work.

**Past and Future Decodability Assumptions.** We now present the key structural assumption of this paper. We assume that a prefix of length $m$ of the history suffices to decode the latent state and also that a suffix of length $n$ of the future suffices to decode the latent state.

Additionally, we will introduce some extra notation for conditioning on either the past or future segments of a sequence. Let $\tilde{o}_{P(h,m)} = \tilde{o}_{\max\{1, h-m\}:h}$ be the past observations and let $\tilde{o}_{F(h,n)} = \tilde{o}_{\min\{1, h+n\}:H}$ refer to the future observations.

**Assumption 1** ($m$-step past decodability). *There exists an* unknown *decoder* $\phi^f_{\star,s} : \tilde{\mathcal{O}}_m \to \mathcal{S}$ *such that for every* reachable *trajectory* $\tau = s_{1:H}$, *we have* $s_h = \phi^f_{\star,s}(\tilde{o}_{P(h,m)})$.

**Assumption 2** ($n$-step future decodability). *There exists an* unknown *decoder* $\phi^b_{\star,s} : \tilde{\mathcal{O}}_n \to \mathcal{S}$ *such that for every* reachable *trajectory* $\tau = s_{1:H}$, *we have* $s_h = \phi^b_{\star,s}(\tilde{o}_{F(h,n)})$.

We note that the decodability assumption on the observation and previous action sequence $\tilde{i}$ is more general than an analogous decodability assumption on the observations alone $o$. Indeed, in practical applications it may be the case that prior actions are required to decode the current state, and hence we work with this more general assumption. In fact, in the experimental section we will show that, empirically, adding actions improves our algorithm's performance.

## 3 PROPOSED OBJECTIVES

In this section, we describe in detail a set of possible inverse kinematic based objectives for the FM-POMDP setting. One is *All History* (AH), which involves using the entire sequence of observations to predict actions. Another is *Forward Jump* (FJ), in which a partial history of the sequence is used from both the past and a number of steps in the future. Finally, *Masked Inverse Kinematics* uses a partial history of the sequence from the past and a partial *future* of the sequence a number of steps in the future. For all of these objectives, we will consider a variant which augments each observation in the input sequence with the previous action. These objectives are visualized in Figure 1 and summarized in Table 1.

Our high-level strategy will be to study which of these objectives are sufficient to obtain a reduction to the analysis in Lamb et al. (2022), which guarantees recovery of the true minimal agent-centric state. To do this, we will first study the Bayes-optimal solution of each objective in terms of the true agent-centric state (section 3.1). Following this, we will study which of these Bayes-optimal solutions are sufficient to complete the reduction in section 3.2.

### 3.1 THE BAYES OPTIMAL CLASSIFIER OF CANDIDATE OBJECTIVES

We start by analyzing the Bayes optimal solution of few inverse kinematics objectives, namely, objectives that aim to predict an action from a sequence of observations. These closed form solutions will later motivate the design of the loss objectives, and guide us towards choosing the proper

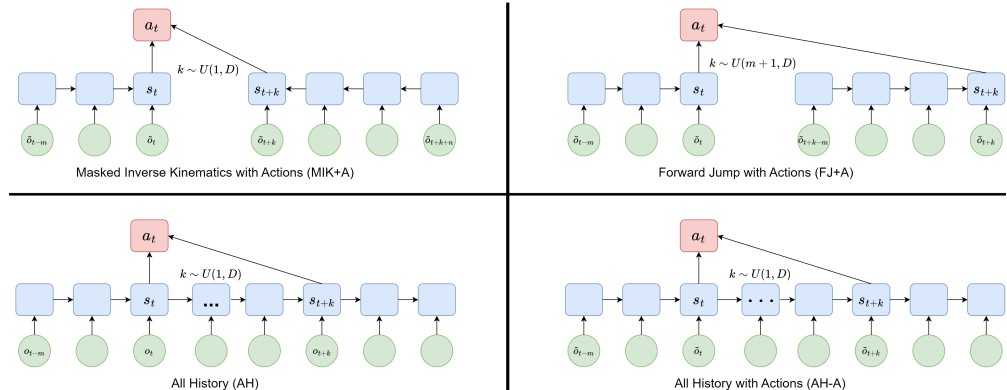

Figure 1: We examine several objectives for generalizing inverse kinematics to FM-POMDPs. MIK+A uses past-decodability and future-decodability with a gap of $k$ masked steps, FJ+A uses past-decodability with a gap of $k$ steps, while AH uses past-decodability over the entire sequence.

way of implementing inverse kinematics for the FM-POMDP setting. These results are proved in Appendix A.2, A.3, A.4.

**Proposed Masked-Inverse Kinematics (MIK+A).** Masked inverse kinematics with actions (MIK+A) achieves the correct Bayes-optimal classifier for the multi-step inverse model, with dependence on only the agent-centric part of the state, i.e., $s_t$ and $s_{t+k}$. Let $s_t = \phi_s^f(\tilde{o}_{\mathrm{P}(t,m)})$, $s_{t+k} = \phi_s^b(\tilde{o}_{\mathrm{F}(t+k,n)})$, $\xi_t = \phi_\xi(\tilde{o}_{1:t})$, $\xi_{t+k} = \phi_\xi(\tilde{o}_{1:t+k})$, $z_t = (s_t, \xi_t)$. Let $k \sim U(1, D)$ . The following result is proved in the appendix for any agent-centric policy $\pi$:

$$\forall k \geq 1, \quad \mathbb{P}_\pi(a_t | \tilde{o}_{\mathrm{P}(t,m)}, \tilde{o}_{\mathrm{F}(t+k,n)}) = \mathbb{P}_\pi(a_t | s_t, s_{t+k}) \tag{1}$$

The MIK objective is essentially the same, except that there is no conditioning on past actions: $\mathbb{P}_\pi(a_t | o_{\mathrm{P}(t,m)}, o_{\mathrm{F}(t+k,n)})$, and would have the same Bayes-optimal classifier result if we relaxed the past and future decodability assumptions to not require actions.

**All History (AH) Objective.** When we condition our encoder on the entire history, the Bayes-optimal multi-step inverse model reduces to a one-step inverse model. Intuitively, the optimal model could simulate an internal one-step inverse model and store these predicted actions in an internal buffer, and then retrieve them as necessary to predict the true actions. The one-step inverse model fails to learn the full agent-centric state, with counterexamples given by Efroni et al. (2022d); Lamb et al. (2022). Let $s_t = \phi_s(o_{1:t})$, $s_{t+k} = \phi_s(o_{1:(t+k)})$, $\xi_t = \phi_\xi(o_{1:t})$, $\xi_{t+k} = \phi_\xi(o_{1:t+k})$, $z_t = (s_t, \xi_t)$. Let $k \sim U(1, D)$. In Appendix A.3, we prove the following:

$$\forall k \geq 1, \quad \mathbb{P}_\pi(a_t | o_{1:t}, o_{1:(t+k)}) = \mathbb{P}_\pi(a_t | s_t, s_{t+1})$$

**All History with actions (AH+A) Objective.** If the observations are augmented with the last action, then these actions can simply be stored to a buffer and retrieved to solve the multi-step inverse modeling problem. Thus the Bayes optimal multi-step inverse model in this setting can have no dependence on the state. In the appendix we prove the following but note that it's a straightforward consequence of this objective conditioning on the action $a_t$ which is being predicted:

$$\forall k \geq 1, \quad \mathbb{P}_\pi(a_t | \tilde{o}_{1:t}, \tilde{o}_{1:(t+k)}) = 1$$

**Forward-Jump Inverse Kinematics (FJ) Objective.** By an almost identical proof as the above, this algorithm achieves the correct Bayes optimal classifier.

$$\forall k, \quad k > m, \quad k \geq 1, \quad \mathbb{P}_\pi(a_t | o_{\mathrm{P}(t,m)}, o_{\mathrm{P}(t+k,m)}) = \mathbb{P}_\pi(a_t | s_t, s_{t+k}). \tag{2}$$

$$\forall k, \quad k \leq m, \quad k \geq 1, \quad \mathbb{P}_\pi(a_t | o_{\mathrm{P}(t,m)}, o_{\mathrm{P}(t+k,m)}) = \mathbb{P}_\pi(a_t | s_t, s_{t+1}). \tag{3}$$

**Forward-Jump Inverse Kinematics with Actions (FJ+A) Objective** Likewise, when conditioning on actions we have:

$$\forall k, \quad k > m, \quad k \geq 1, \quad \mathbb{P}_\pi(a_t | \tilde{o}_{\mathrm{P}(t,m)}, \tilde{o}_{\mathrm{P}(t+k,m)}) = \mathbb{P}_\pi(a_t | s_t, s_{t+k}). \tag{4}$$

$$\forall k, \quad k \leq m, \quad k \geq 1, \quad \mathbb{P}_\pi(a_t | \tilde{o}_{\mathrm{P}(t,m)}, \tilde{o}_{\mathrm{P}(t+k,m)}) = 1. \tag{5}$$

## 3.2    DISCOVERING THE COMPLETE AGENT-CENTRIC STATE

In previous section we described several inverse kinematic terms that may be useful for discovering the agent-centric state representation of an FM-POMDP. We now claim that among this set of inverse kinematics terms, the MIK+A is the most favorable one: the main result from Lamb et al. (2022) (Theorem 5.1) implies that MIK+A recovers the agent-centric state representation. Further, we elaborate on the failure of the other inverse kinematic objectives.

**MIK+A Discovers the Full Agent-Centric State.** Given successful recovery of the Bayes optimal classifier for the multi-step inverse model, with dependence on only $s_t$ and $s_{t+k}$, we can reuse the theory from (Lamb et al., 2022), with slight modifications, as given in the appendix. The most important modification is that we roll-out for $m+n+D$ steps, to ensure that we have enough steps to decode $s_t$ and $s_{t+k}$, where $D$ is the diameter of the agent-centric state. With the above, the reduction to Theorem 5.1 of Lamb et al. (2022) is natural. There, the authors showed that, under proper assumptions, if an encoder $\phi$ can represent inverse kinematic terms of the form $\mathbb{P}_\pi(a_t|s_t, s_{t+k})$ for all $k \in [D]$ then $\phi$ is the mapping from observations to the agent-centric state.

**Failure of All History (AH) for Discovering Agent-Centric State Representation.** We showed that the AH objective can be satisfied by only solving the one-step inverse objective $p(a_t|s_t, s_{t+1})$. It was shown in (Rakelly et al., 2021; Lamb et al., 2022; Efroni et al., 2022d) that the one-step inverse objective learns an undercomplete representation. Intuitively, it may incorrectly merge states which have locally similar dynamics but are actually far apart in the environment.

**Failure of Forward-Jump (FJ and FJ+A) for Discovering Agent-Centric State Representation.** Since the Forward-Jump objectives only rely on past-decodability, it does not have correct Bayes optimal classifiers for all $k \leq m$. Namely, it does not recover the inverse model with $k$ in this regime. This prevents us from applying the result of Lamb et al. (2022), since it requires the set of all inverse models $k \in [D]$, wheres FJ only has access to $k \in \{1, m, m+1, .., D\}$ but not for $k$ in intermediate values.

Nevertheless, this give rise on an intriguing question: is there a counterexample that shows FJ or FJ+A does not work? We establish a counterexample in which the $k = 1$ examples are insufficient to distinguish all of the states and where the $k > 3$ examples are useless. We will then construct an observation space for an FM-POMDP with $m = 3$, which will then cause both the FJ and FJ+A objectives to fail.

Consider the following agent-centric state with two components $s = (s^A, s^B)$. $s^A$ receives four values $\{0, 1, 2, 3\}$ and follows the dynamics $s_{t+1}^A = (s_t^A + a_t) \bmod 4$, which is a simple cycle with a period of 4, controlled by the action $a \in \{0, 1\}$. $s^B = a_{t-1}$ simply records the previous action. We have an exogenous periodic signal $c_{t+1} = (c_t + 1) \bmod 4$. This FM-POMDP's agent-centric state has a diameter of $D = 3$, and the true state can be recovered with $k$ from 1 to 3. However, all multi-

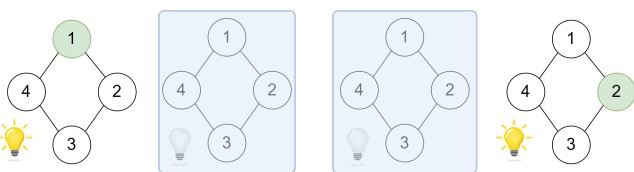

Figure 2: The Forward Jump objective fails in a counterexample where the observation can only be seen once every $m$ steps, preventing the use of $k \leq m$ inverse kinematics examples, whereas the inverse examples with $k > m$ provide no signal for separating the states.

step inverse problems, under the random policy, with $k > 3$ has the same probability of $0.5$ for both actions. Concretely, for any plan to reach a goal with $k > 3$ steps, multiplying the actions by -1 will still yield an equally optimal plan with respect to $s^A$, while only the last action taken has an effect on $s^B$, so the distribution over the first action will be uniform (MDP shown in appendix figure 8). Now, let's turn to the construction of the observation space of this FM-POMDP. We will use the counter $c_t$ to control when the state can be seen in the observation, so if $c_t = 0$, we have $o_t = s_t$, whereas if $c_t \neq 0$, we have $o_t = -1$ (blank observation). It is apparent that if $c_t \neq 0$, that we can't decode the state from the current observation. However, with a history of $m = 3$ past observations, we can decode the state by finding $s_t$ when it is present in the observation (i.e. when $o_t \neq -1$), and then simulate the last $m$ steps using the previous actions recorded in the observations. A simplified version of this construction (showing only $s^A$ and with $m = 2$) is shown in Figure 2.

| Methods | Objective | Correct Bayes Optimal Classifier | Complete Agent-Centric State | Assumes Past Decodability | Assumes Future Decodability | Discards Exogenous Noise |
|---|---|---|---|---|---|---|
| AH | $\mathbb{P}_\pi(a_t\vert o_{1:t}, o_{1:(t+k)})$ | ✗ | ✗ | ✓ | ✗ | ✓ |
| AH+A | $\mathbb{P}_\pi(a_t\vert \tilde{o}_{1:t}, \tilde{o}_{1:(t+k)})$ | ✗ | ✗ | ✓ | ✗ | ✓ |
| FJ | $\mathbb{P}_\pi(a_t\vert o_{\mathrm{P}(t,m)}, o_{\mathrm{P}(t+k,m)})$ | ✓ | ✗ | ✓ | ✗ | ✓ |
| FJ+A | $\mathbb{P}_\pi(a_t\vert \tilde{o}_{\mathrm{P}(t,m)}, \tilde{o}_{\mathrm{P}(t+k,m)})$ | ✓ | ✗ | ✓ | ✗ | ✓ |
| MIK | $\mathbb{P}_\pi(a_t\vert \tilde{o}_{\mathrm{P}(t,m)}, \tilde{o}_{\mathrm{F}(t+k,n)})$ | ✓ | ✗ | ✓ | ✓ | ✓ |
| MIK+A | $\mathbb{P}_\pi(a_t\vert \tilde{o}_{\mathrm{P}(t,m)}, \tilde{o}_{\mathrm{F}(t+k,n)})$ | ✓ | ✓ | ✓ | ✓ | ✓ |

Table 1: A summary of the baseline inverse kinematics approaches which we study. Our final proposed method Masked Inverse Kinematics with actions (MIK+A) has a significant advantages over the alternatives: it can provably recover the agent centric state representation.

To reiterate the claim of the proof, we constructed a FM-POMDP where it is necessary to use $k = 2$ and $k = 3$ multi-step model examples to separate out the states correctly. Yet the state can only be perfectly decoded with $m = 3$ steps of history. Thus, the FJ and FJ+A objectives fail to learn the correct representation in this FM-POMDP.

## 4 EXPERIMENTAL RESULTS

We experimentally validate whether the set of inverse kinematic based objectives can recover the agent-centric state in the FM-POMDP setting. To do this, we first evaluate the objectives in a partially observable navigation environment (section 4.1) and then study whether these objectives can learn useful representations, in presence of partially observable offline datasets (section 4.2).

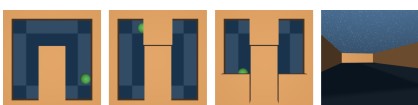

Figure 3: Visualization of the four navigation environments. From left to right: no curtain, one curtain, three curtains, and first-person environments. All include some degree of partial observability.

### 4.1 DISCOVERING STATE FROM PARTIALLY-OBSERVED NAVIGATION ENVIRONMENTS

**Experiment Setup** We first consider the navigation environments in Figure 3, with other figures and details in Appendix figures 9 and 10, and introduce partial observability in these tasks. Details on experimental setup are provided in appendix D.1. In this problem, m-step past decodability is achieved with m=1. The n-step future decodability assumption subtly violated in cases where the agent collides into a wall and loses all of its velocity. The agent's velocity before hitting the wall is then not decodable from any number of future observations. We also consider an optional Self-Prediction (SP) objective $\Vert sg(s_{t+k}) - f(s_t, k)$, where sg refers to stopping gradients. This auxiliary objective, inspired by Guo et al. (2022); Tang et al. (2023) can help to improve the quality of representations.

**Experiment Results** In the acceleration-control experiments (Figure 4, Table 2), we consistently found that MIK+A has the best performance, which is aligned with theory. The theory also suggests that AH+A has no state dependence, and we indeed see that it has the worst performance, when the maximum $k$ is small. Another striking result is that AH with any maximum $k$ is theoretically equivalent to AH with maximum $k$ of 1, and these two methods indeed have very similar errors experimentally. Further evidence comes from investigating the action prediction losses (Table 5), where we see that AH+A has nearly zero error while AH has a very low loss, supporting

| Objective | No Curtain | Three Curtains |
|---|---|---|
| No History | 47.6 | 52.7 |
| AH | 9.9 | 13.2 |
| AH+A | 18.8 | 18.9 |
| FJ | 10.0 | 15.3 |
| FJ+A | 5.8 | 7.2 |
| MIK | 10.1 | 14.7 |
| MIK+A | 6.1 | 7.4 |

Table 2: State Estimation Errors (%) on various tasks with exogenous noise.

our claim that these objectives fail because they reduce the bayes optimal predictor to an overly simple learning objective. Another finding is that FJ+A and MIK+A are fairly similar, which suggests that the theoretical counterexample for FJ+A may not imply poor performance. Extra experiment results of adding next-state prediction or exogenous noise are provided in appendix D.1.

### 4.2 VISUAL OFFLINE RL WITH PARTIAL OBSERVABILITY

We validate the proposed objectives in challenging pixel-based visual offline RL tasks, using the vd4rl benchmark dataset Lu et al. (2022). For our experiments, we follow the same setup as Islam

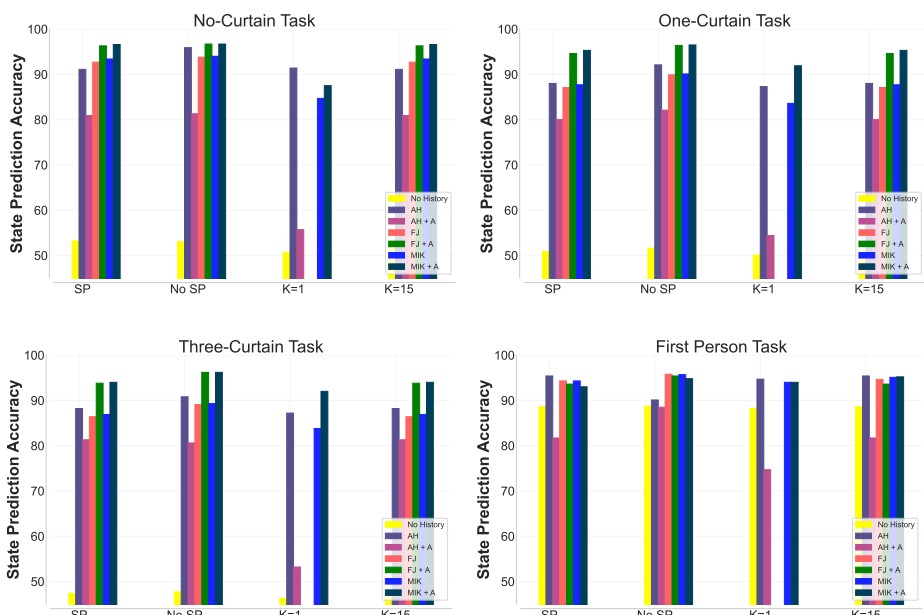

Figure 4: We compare state estimation performance (higher is better) across our various proposed methods. We compare action-conditioned and action-free variants while also considering a self-prediction auxiliary loss and the maximum prediction span $K$. We omit FJ and FJ+A in the maximum $K = 1$ case because of equivalence to AH and AH+A with a shorter history.

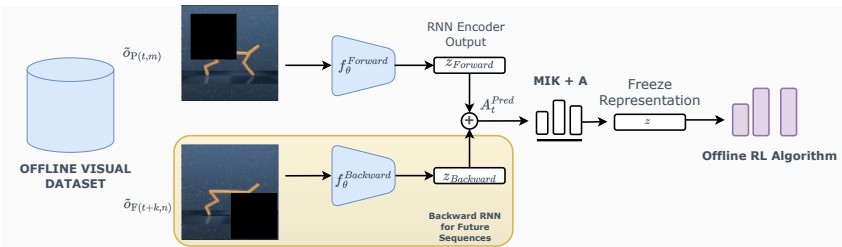

Figure 5: **Illustration of the visual offline RL experiment setup, in presence of partial observability.** We use a forward and backward sequence model (RNN encoder) to handle past and future observation sequences, to achieve latent state discovery in FM-POMDPs.

et al. (2023), where we pre-train the representations from the visual observations and then perform fine-tuning on the fixed representations using the TD3+BC offline RL algorithm. In our experiments, we compare results using several variations of our proposed objectives, along with several other baselines. We mainly compare with five other baselines, namely ACRO (Islam et al., 2023), DRIML (Mazoure et al., 2020), HOMER (Misra et al., 2020), CURL (Laskin et al., 2020) and 1-step inverse action prediction (Pathak et al., 2017a).

**Experiment Setup :** We consider an offline RL setting with partial observations, as illustrated in figure 5. To do this, we use the existing vd4rl benchmark dataset Lu et al. (2022), and to turn it into a POMDP setting, we apply masking or patching on the observation space randomly. In other words, for each batch of samples from the offline dataset, we randomly patch each observation with a masking of size $16 \times 16$ to make the observations partially observable to the model. In addition to that, since existing Lu et al. (2022) setup uses pixel-based observations and uses a framestacking of 3, to make the setting even more challenging, we randomly zero out 2 out of 3 stacked frames. We do this so that the model can only see both the stacked frames and each frame partially; with the goal to see if our proposed objectives using a forward and backward running sequence model can be more robust with the learnt representations.

**Experiment Results :** Our experimental results show that in presence of partial observability, most of the existing baselines as in Islam et al. (2023) can fail considerably, for different domains and

datasets. In contrast, when we consider the history information and also additionally take into account the action information, then performance of the proposed models can improve significantly. Note that all our experiments here only consider the pixel-based datasets from Lu et al. (2022) with only adding partial observability, without considering any exogenous noise in the datasets as in the setups in Islam et al. (2023). Figure 6 shows that in presence of partial observability, all the considered baselines can fail considerably and performance degrades significantly compared to what was reported in the fully observed setting. In comparison, the proposed objectives can be more robust in partial observability, and notably our key objective (MIK+A) can perform significantly compared to other model ablations. Experimental results show that **MIK + A** can perform significantly better comopared to baselines, in almost all of the tasks. Figure 7 shows results for an even more difficult experiment setup with randomly zeroing stacked frames. Experimental results show that **MIK + A** can still perform relatively better compared to other baselines, in this difficult setting, since the forward and backward sequence models capturing the past and future observations can better capture sufficient information from the partial observations to fully recover the agent-centric state.

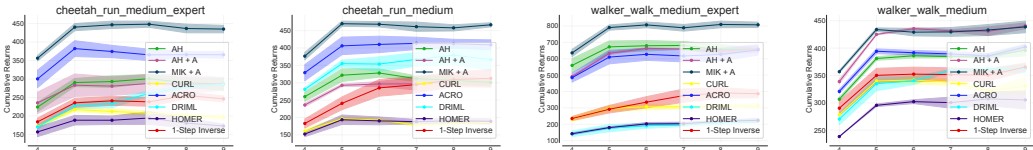

Figure 6: Visual offline datasets from Lu et al. (2022) with **patching** ($16 \times 16$) **to make the observations partially observable.** We compare several of the proposed objectives discussed earlier, along with few baselines, using the representation learning setup in Islam et al. (2023). Experimental results are compared across 3 different domains (Cheetah-Run, Walker-Walk and Humanoid-Walk) and 2 different datasets (Expert and Medium-Expert), across 5 different random seeds.

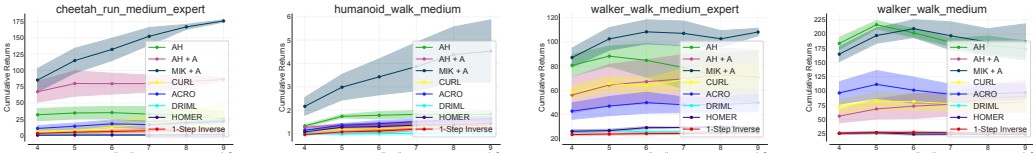

Figure 7: A more challenging setting where in addition to the patching of observations, we further apply **randomly zeroing of frame-stacking**. We apply framestacking for visual observations, where to make the task more difficult and partially observable, we randomly *zero out 2 out of 3 frames*, on top of the masked observations.

## 5 RELATED WORK

Our work builds up on two closely related line of work : (a) on short-term memory POMDPs and (b) learning agent-centric latent states. We describe closely related work on partially observable settings, both theoretical and empirical, and discuss why existing works fail to fully recover the agent-centric latent state in a partially observed setting.

**Theoretical Research on FM-POMDPs.** Efroni et al. (2022b); Liu et al. (2022); Zhan et al. (2022); Wang et al. (2022a) studied finite-sample guarantees under closely related m-step past and n-step future decodability assumptions. Nevertheless, their algorithms are currently impossible to scale and implement with standard oracles (such as log-likelihood minimization) since it requires an optimistic optimization over a set of functions .Further, unlike our reward-free setting, their algorithm is dependent on having a reward signal, whereas our work focuses on reward-free representation learning. Lastly, these works did not considered the high-dimensional problem in the presence of exogenous and time correlated noise.

**Empirical Research on POMDPs.** Partial observability is a central challenge in practical reinforcement learning settings and, as such, it has been the focus of a large body of empirical work. Seminal large scale empirical deep RL research has considered serious partial observability, such as the OpenAI Five program for Dota 2 (OpenAI et al., 2019) and the DeepMind AlphaStar system (Mathieu et al., 2023). Much of this work has used a recurrent neural network or other sequence model to handle a state with history. While much of this work is focused on directly learning a policy or value function (Hausknecht & Stone, 2017), these approaches will fail when reward is absent. Other

work has learned a recurrent forward model to predict observations as well (Igl et al., 2018; Hafner et al., 2019b; 2020), yet this will fail when exogenous noise is dominant. To our knowledge, none of these DeepRL POMDP works have considered our proposed setting of learning agent-centric state with inverse kinematics. Ni et al. (2022) showed an extensive empirical benchmark where recurrent online RL is used for POMDPs. This differs from our work principally in that it's empirical and focused on reward-signal, whereas our approach is reward-free and the motivation for our loss objectives is a consequence of asymptotic theory we develop.

**Research on Agent-Centric States and Inverse Kinematics.** The primary line of theoretical research on inverse kinematics and agent-centric states is exclusively concerned with the MDP setting (Lamb et al., 2022; Efroni et al., 2022a;c;d; Islam et al., 2023; Mhammedi et al., 2023; Hutter & Hansen, 2022). In particular, much of this work has focused on analysis showing that the agent-centric state can be provably recovered under some assumptions. The PPE method (Efroni et al., 2022d) introduced multi-step inverse kinematics in the deterministic dynamics, episodic setting with fixed start states. Lamb et al. (2022) extended this to the non-episodic setting, while Efroni et al. (2022a) handles a stochastic dynamics setting. Tomar et al. (2023); Islam et al. (2023) considered multi-step inverse models for offline-RL, while only considering the fully-observed setting. While Brandfonbrener et al. (2023) used pre-trained multi-step and one-step inverse models for online RL, still in the fully-observed setting. Pathak et al. (2017b); Shelhamer et al. (2017); Badia et al. (2020); Schmeckpeper et al. (2020); Rakelly et al. (2021) all use one-step inverse objective in fully-observed setting to improve empirical performance. Bharadhwaj et al. (2022) InfoPower used a one-step inverse objective along with an RNN to encode the history. Wang et al. (2022c) showed discovery of agent-centric state using causal independence tests and was restricted to the fully-observed setting. Wang et al. (2022b) studied learning a recurrent forward model with a factorization of the state space into agent-centric and exogenous components. This method naturally handles POMDPs, but requires learning both the agent-centric and exogenous states to satisfy the future observation prediction objective, so differs significantly from our algorithmic approach, that allows to directly avoid learning information on the exogenous noise.

**Work related to both POMDPs and Multi-step Inverse Kinematics.** To our knowledge, ours is the first work to explicitly consider inverse kinematics for learning agent-centric states in the POMDP setting. Our counter-examples to AH and AH+A objectives, where the model can fail to learn the state by memorizing actions, is reminiscent of the causal confusion for imitation learning work (De Haan et al., 2019) . Baker et al. (2022) considers a one-step inverse model using a transformer encoder, to learn an action-labeling model. While this is equivalent to our All History (AH) approach, the focus of that work was not on learning representations. Sun et al. (2023); Goyal et al. (2022) consider a sequence learning setup where a bidirectional sequence model masks observations and actions in the input and predicts the masked actions. While these approaches seem consistent with our theoretical analysis, they use a bidirectional model and therefore learn an entangled model of $\phi_s^f$ and $\phi_s^b$ in their internal representations, where the correct usage for planning and exploration is unclear. This makes their setting different from our focus on learning an explicit state representation and their work doesn't provide a theoretical analysis,

## 6 CONCLUSION

Partially observable settings in RL are often difficult to work with, theoretically without strong assumptions, and empirically with a implementable algorithm, despite the generality of non-Markovian observations that can arise naturally in practice. To recover the agent-centric full latent state that can be considered as an information state, is quite difficult in the FM-POMDP setting. Several works using multi-step inverse kinematics has recently been proposed for latent state discovery, in the theoretical and empirical RL communities. However, despite the popularity, how to apply multi-step inverse kinematics in the FM-POMDP setting has not been previously studied. Our work shows that it's possible to succeed in discovering agent-centric states in FM-POMDPs while many intuitive algorithms fail. We made the assumptions of past-decodability (Efroni et al., 2022b) while introducing a new future-decodability assumption. In this work, we demonstrated several examples showing that the full agent-centric state can be recovered from partially observable, offline pre-collected data for acceleration and control. Additionally, we showed that MIK+A, taking the action information from past and future into account, can be effective for learning a latent representation that can improve performance empirically on a challenging partially observable offline RL task. A natural topic for future work is developing an online algorithm which discovers a policy that achieves these decodability properties rather than assuming them.

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

# Appendix

In the appendix, we include proofs and counterexamples for our theoretical results, and environment details and additional results for the experimental setup.

## A  THEORY DETAILS

### A.1  STRUCTURAL LEMMA

We now describe a structural result of the agent-centric FM-POMDP model, closely following the proof in Efroni et al. (2022d). We say that $\pi$ is an *agent-centric policy* if it is not a function of the exogenous noise. Formally, for any history of action-augmented observations $o_1$ and $o_2$, if $\phi^f_{\star,s}(o_1) = \phi^f_{\star,s}(o_2)$ then $\pi(\cdot \mid o_1) = \pi(\cdot \mid \phi^f_{\star,s}(o_2))$. Let $\mathbb{P}_\pi(s' \mid s, h)$ be the probability to observe the control-endogenous latent state $s = \phi^f_{\star,s}(\tilde{o}_{\mathrm{P}(h,m)})$, $h$ time steps after observing step $t$, $s' = \phi^b_{\star,s}(\tilde{o}_{\mathrm{P}(t+h,m)})$ and following policy $\pi$, starting with action $a$. Note that the claim will also hold if $s$ or $s'$ use either the forward or the backward encoder. Let the exogenous state be defined similarly as $\xi = \phi^\star_\xi(\tilde{o}_{1:t})$ and $\xi' = \phi^\star_\xi(\tilde{o}_{1:t+h})$. The following result shows that, when executing an endogenous policy, the future $h$ time step distribution of the observation process conditioning on any $o$ has a decoupling property.

**Lemma 1** (Decoupling property for endogenous policies Efroni et al. (2022d)). *Let $\mu$ be the initial distribution. Assume that the agent-centric and exogenous part is decoupled for the initial distribution, $\mu(s, \xi) = \mu(s)\mu(\xi)$, and that $\pi$ is an endogenous policy. Then, for any $t \geq 1$ it holds that $\mathbb{P}_\pi(o' \mid o, a, h) = q(o' \mid s', \xi')\mathbb{P}_\pi(s' \mid s, a, h)\mathbb{P}(\xi' \mid \xi, h)$.*

This lemma is a key result for analyzing the Bayes optimal solution of the different inverse kinematic objectives described in this work. We assume that the policy only depends on the state so that it rules out the known hard problems with unobserved confounders. Some recent work in causal inference literature mitigate the unobserved confounding issue by integrating the offline and online datasets (Wu & Yang, 2022; Cheng et al., 2023).

## A.2 Proof that the MIK+A objective has the right Bayes optimal classifier

We analyze the Bayes optimal classifier of the MIK+A objective, by first applying Bayes theorem on the action $a_t$ and the future observation sequence starting from $t+k$. We then use the decodability assumption to introduce the latent variable $z_{t+k}$ and then apply the Markov assumption on the latent space on step $t + k$. We then cancel the probability over future observations because it has no action dependence and also cancel the exogenous-noise part of the latent state.

We use the following notation for all $t$:

$$s_t = \phi_s^f(\tilde{o}_{P(t,m)})$$
$$s_{t+k} = \phi_s^b(\tilde{o}_{F(t+k,n)})$$
$$\xi_t = \phi_\xi(\tilde{o}_{1:t})$$
$$\xi_{t+k} = \phi_\xi(\tilde{o}_{1:t+k})$$
$$z_t = (s_t, \xi_t).$$

We have that for all $k > t$:

$$
\begin{aligned}
\mathbb{P}_\pi(a_t | \tilde{o}_{P(t,m)}, \tilde{o}_{F(t+k,n)}) &= \frac{\mathbb{P}_\pi(\tilde{o}_{F(t+k,n)} | \tilde{o}_{m(t)}, a_t) \pi(a_t | \tilde{o}_{P(t,m)})}{\sum_{a'} \mathbb{P}_\pi(\tilde{o}_{F(t+k,n)} | \tilde{o}_{P(t,m)}, a') \pi(a' | \tilde{o}_{P(t,m)})} \\
&= \frac{\mathbb{P}_\pi(\tilde{o}_{F(t+k,n)} | z_t, a_t) \pi(a_t | z_t)}{\sum_{a'} \mathbb{P}_\pi(\tilde{o}_{F(t+k,n)} | z_t, a') \pi(a' | z_t)} \\
&= \frac{\mathbb{P}_\pi(\tilde{o}_{F(t+k,n)}, z_{t+k} | z_t, a_t) \pi(a_t | z_t)}{\sum_{a'} \mathbb{P}_\pi(\tilde{o}_{F(t+k,n)}, z_{t+k} | z_t, a') \pi(a' | z_t)} \\
&= \frac{\mathbb{P}_\pi(\tilde{o}_{F(t+k,n)} | z_{t+k}) \mathbb{P}_\pi(z_{t+k} | z_t, a_t) \pi(a_t | z_t)}{\sum_{a'} \mathbb{P}_\pi(\tilde{o}_{F(t+k,n)} | z_{t+k}) p(z_{t+k} | z_t, a') \pi(a' | z_t)} \\
&= \frac{\mathbb{P}_\pi(z_{t+k} | z_t, a_t) \pi(a_t | z_t)}{\sum_{a'} \mathbb{P}_\pi(z_{t+k} | z_t, a') \pi(a' | z_t)} \\
&= \frac{\mathbb{P}_\pi(s_{t+k} | s_t, a_t) \mathbb{P}(\xi_{t+k} | \xi_t) \pi(a_t | s_t)}{\sum_{a'} \mathbb{P}_\pi(s_{t+k} | s_t, a') \mathbb{P}(\xi_{t+k} | \xi_t) \pi(a' | s_t)} \\
&= \frac{\mathbb{P}_\pi(s_{t+k} | s_t, a_t) \pi(a_t | s_t)}{\sum_{a'} \mathbb{P}_\pi(s_{t+k} | s_t, a') \pi(a' | s_t)} \\
&= \mathbb{P}_\pi(a_t | s_t, s_{t+k}).
\end{aligned}
$$

The first relation holds by Bayes rule. The third relation holds since $z_{t+k}$ is a deterministic function of $\tilde{o}_{F(t+k,n)}$ under the future decodability assumption. The forth relation holds by Bayes rule, along with the assumption that the future observations following $t + k$ are conditionally independent with $(z_t, a_t)$ given $z_{t+k}$ (since we assume $\pi$ on time step $t$ only depends on $s_t$, since it's an agent-centric policy). The sixth relation holds by the decoupling lemma, Lemma 1.

## A.3 Proof that All-History (AH) reduces to one-step inverse model

We analyze the Bayes-optimal classifier of the AH objective. We first apply Bayes theorem between the observation sequence and the predicted action $a_t$. We then use the past-decodability assumption to introduce latent variables $z_t$ and $z_{t+1}$. We apply the chain rule of probability and then markov independence of the observations given the latent states. The observations conditioned on the latents then cancel, and then the exogenous noise dynamics also cancel, leaving the one-step inverse model over the latent states.

We use the following notation for all $t$.

$$s_t = \phi_s^f(o_{1:t})$$
$$s_{t+k} = \phi_s^f(o_{1:(t+k)})$$
$$\xi_t = \phi_\xi(o_{1:t})$$
$$\xi_{t+k} = \phi_\xi(o_{1:t+k})$$
$$z_t = (s_t, \xi_t)$$

The following relations hold.

$$
\begin{aligned}
\mathbb{P}(a_t|o_{1:t}, o_{1:t+k}) &= \mathbb{P}(a_t|o_{1:t+k}) \\
&= \frac{\mathbb{P}(o_{1:t+k}|a_t)\mathbb{P}(a_t)}{\sum_{a'}\mathbb{P}(o_{1:t+k}|a')\mathbb{P}(a')} \\
&= \frac{\mathbb{P}(o_{1:t}, o_{(t+1):t+k}, z_t, z_{t+1}|a_t)\mathbb{P}(a_t)}{\sum_{a'}\mathbb{P}(o_{1:t}, o_{(t+1):t+k}, z_t, z_{t+1}|a')\mathbb{P}(a')} \\
&= \frac{\mathbb{P}(o_{1:t}|a_t)\mathbb{P}(z_t|o_{1:t}, a_t)\mathbb{P}(z_{t+1}|o_{1:t}, z_t, a_t)\mathbb{P}(o_{(t+1):t+k}|o_{1:t}, z_t, z_{t+1}, a_t)\mathbb{P}(a_t)}{\sum_{a'}\mathbb{P}(o_{1:t}|a')\mathbb{P}(z_t|o_{1:t}, a')\mathbb{P}(z_{t+1}|o_{1:t}, z_t, a')\mathbb{P}(o_{(t+1):t+k}|o_{1:t}, z_t, z_{t+1}, a')\mathbb{P}(a')} \\
&= \frac{\mathbb{P}(o_{1:t}|a_t)\mathbb{P}(z_t|o_{1:t})\mathbb{P}(z_{t+1}|z_t, a_t)\mathbb{P}(o_{(t+1):t+k}|z_{t+1})\mathbb{P}(a_t)}{\sum_{a'}\mathbb{P}(o_{1:t}|a')\mathbb{P}(z_t|o_{1:t})\mathbb{P}(z_{t+1}|z_t, a')\mathbb{P}(o_{(t+1):t+k}|z_{t+1})\mathbb{P}(a')} \\
&= \frac{\mathbb{P}(o_{1:t})\mathbb{P}(z_t|o_{1:t})\mathbb{P}(z_{t+1}|z_t, a_t)\mathbb{P}(o_{(t+1):t+k}|z_{t+1})\pi(a_t|o_{1:t})}{\sum_{a'}\mathbb{P}(o_{1:t})\mathbb{P}(z_t|o_{1:t})\mathbb{P}(z_{t+1}|z_t, a')\mathbb{P}(o_{(t+1):t+k}|z_{t+1})\pi(a'|o_{1:t})} \\
&= \frac{\mathbb{P}(o_{1:t})\mathbb{P}(z_t|o_{1:t})\mathbb{P}(z_{t+1}|z_t, a_t)\mathbb{P}(o_{(t+1):t+k}|z_{t+1})\pi(a_t|s_t)}{\sum_{a'}\mathbb{P}(o_{1:t})\mathbb{P}(z_t|o_{1:t})\mathbb{P}(z_{t+1}|z_t, a')\mathbb{P}(o_{(t+1):t+k}|z_{t+1})\pi(a'|s_t)} \\
&= \frac{\mathbb{P}(z_{t+1}|z_t, a_t)\pi(a_t|s_t)}{\sum_{a'}\mathbb{P}(z_{t+1}|z_t, a')\pi(a'|s_t)} \\
&= \frac{\mathbb{P}(s_{t+1}|s_t, a_t)\mathbb{P}(\xi_{t+1}|\xi_t)\pi(a_t|s_t)}{\sum_{a'}\mathbb{P}(s_{t+1}|s_t, a')\mathbb{P}(\xi_{t+1}|\xi_t)\pi(a'|s_t)} \\
&= \frac{\mathbb{P}(s_{t+1}|s_t, a_t)\pi(a_t|s_t)}{\sum_{a'}\mathbb{P}(s_{t+1}|s_t, a')\pi(a'|s_t)} \\
&= \mathbb{P}_\pi(a_t|s_t, s_{t+1}).
\end{aligned}
$$

The second relation holds by Bayes rule. The third relation holds by since $z_t$ and $z_{t+1}$ are deterministic functions of the observation sequence by the decodability assumptions. The sixth relation holds by using:

$$\mathbb{P}(o_{1:t} \mid a_t)\mathbb{P}(a_t) = \mathbb{P}(o_{1:t})\pi(a_t \mid o_{1:t})$$

due to Bayes rule. The seventh relation holds by the fact we assume $\pi$ is agent centric. The ninth relation holds by the decoupling lemma, Lemma 1.

### A.4   PROOF THAT ALL-HISTORY WITH ACTIONS (AH+A) HAS NO STATE DEPENDENCE

The Bayes-optimal classifier for the AH+A objective can be perfectly satisfied without using the state, by simply memorizing the sequence of actions and retrieving them. Note that in practice, this is easiest to achieve when the maximum $K$ value is small.

The following relations holds for all $k > t$ since there is an explicit conditioning in the probability distribution.

$$\mathbb{P}(a_t|x_{1:t}, a_{1:t}, x_{1:t+k}, a_{1:t+k}) = \mathbb{P}(a_t|a_{1:t+k}) = \mathbb{P}(a_t|a_t) = 1.$$

This implies the AH+A Bayes solution erases the information on the agent-centric state, since the action can be directly predicted.

## B    FORWARD JUMP COUNTEREXAMPLE

Figure 8 contains a counter-example for the forward-jump (FJ) objective, for why FJ fails to capture the agent-centric state in partial observability.

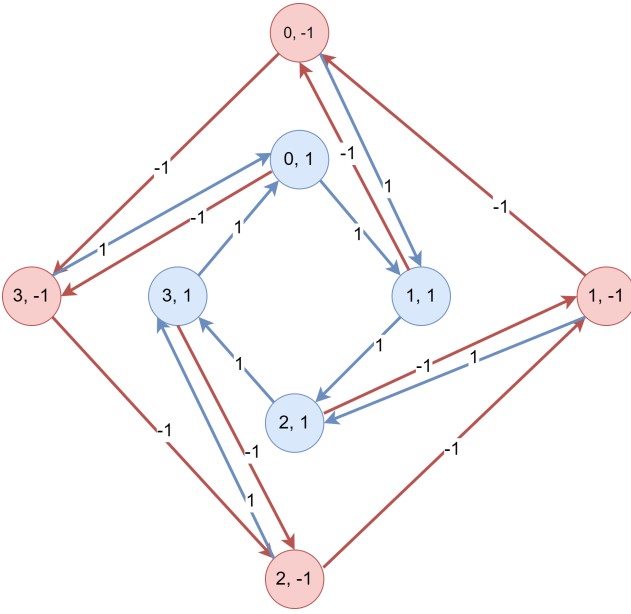

Figure 8: The full underlying MDP of the counterexample for the Forward Jump objective. Each of the eight states shows $(s^A, s^B)$, with the coloring used to reflect $s^B$ and the action which can reach it. The special property of this MDP is that its multi-step inverse model examples with $k \geq 4$ are uninformative while its $k = 1$ examples are insufficient. This creates a counterexample for methods solely relying on past-decodability, because the length of history required to decode the state may overlap and prevent access to the $k = 2$ and $k = 3$ inverse kinematics examples.

With a small computer program we generated the inverse kinematics examples for this MDP from $k = 1$ to $k = 10$. First, we generated the 16368 inverse kinematic examples and verified that examples with $4 \leq k \leq 10$ have a uniform distribution for the first action.

All of the inverse kinematics examples up to $k = 6$ are included in a text file in the supplementary materials. A subset where the initial state is either $(0, -1)$ or $(0, 1)$ is shown below to give a flavor of the structure, in which $k = 4$ has uniform probability over the first action whereas $k = 2$ has more useful inverse kinematic examples:

$k = 2$ examples:
$(0, \text{-}1) \rightarrow (0, -1)$    via    a:(1, -1)
$(0, -1) \rightarrow (0, 1)$    via    a:(-1, 1)
$(0, -1) \rightarrow (2, -1)$    via    a:(-1, -1)
$(0, -1) \rightarrow (2, 1)$    via    a:(1, 1)
$(0, 1) \rightarrow (0, -1)$    via    a:(1, -1)
$(0, 1) \rightarrow (0, 1)$    via    a:(-1, 1)
$(0, 1) \rightarrow (2, -1)$    via    a:(-1, -1)
$(0, 1) \rightarrow (2, 1)$    via    a:(1, 1)

$k = 4$ examples:
$(0, \text{-}1) \rightarrow (0, -1)$    via    a:(-1, -1, -1, -1)
$(0, -1) \rightarrow (0, -1)$    via    a:(-1, 1, 1, -1)
$(0, -1) \rightarrow (0, -1)$    via    a:(1, -1, 1, -1)
$(0, -1) \rightarrow (0, -1)$    via    a:(1, 1, -1, -1)
$(0, -1) \rightarrow (0, 1)$    via    a:(-1, -1, 1, 1)

$(0,-1) \rightarrow (0,1)$   via   a:(-1, 1, -1, 1)
$(0,-1) \rightarrow (0,1)$   via   a:(1, -1, -1, 1)
$(0,-1) \rightarrow (0,1)$   via   a:(1, 1, 1, 1)
$(0,-1) \rightarrow (2,-1)$   via   a:(-1, -1, 1, -1)
$(0,-1) \rightarrow (2,-1)$   via   a:(-1, 1, -1, -1)
$(0,-1) \rightarrow (2,-1)$   via   a:(1, -1, -1, -1)
$(0,-1) \rightarrow (2,-1)$   via   a:(1, 1, 1, -1)
$(0,-1) \rightarrow (2,1)$   via   a:(-1, -1, -1, 1)
$(0,-1) \rightarrow (2,1)$   via   a:(-1, 1, 1, 1)
$(0,-1) \rightarrow (2,1)$   via   a:(1, -1, 1, 1)
$(0,-1) \rightarrow (2,1)$   via   a:(1, 1, -1, 1)
$(0,1) \rightarrow (0,-1)$   via   a:(-1, -1, -1, -1)
$(0,1) \rightarrow (0,-1)$   via   a:(-1, 1, 1, -1)
$(0,1) \rightarrow (0,-1)$   via   a:(1, -1, 1, -1)
$(0,1) \rightarrow (0,-1)$   via   a:(1, 1, -1, -1)
$(0,1) \rightarrow (0,1)$   via   a:(-1, -1, 1, 1)
$(0,1) \rightarrow (0,1)$   via   a:(-1, 1, -1, 1)
$(0,1) \rightarrow (0,1)$   via   a:(1, -1, -1, 1)
$(0,1) \rightarrow (0,1)$   via   a:(1, 1, 1, 1)
$(0,1) \rightarrow (2,-1)$   via   a:(-1, -1, 1, -1)
$(0,1) \rightarrow (2,-1)$   via   a:(-1, 1, -1, -1)
$(0,1) \rightarrow (2,-1)$   via   a:(1, -1, -1, -1)
$(0,1) \rightarrow (2,-1)$   via   a:(1, 1, 1, -1)
$(0,1) \rightarrow (2,1)$   via   a:(-1, -1, -1, 1)
$(0,1) \rightarrow (2,1)$   via   a:(-1, 1, 1, 1)
$(0,1) \rightarrow (2,1)$   via   a:(1, -1, 1, 1)
$(0,1) \rightarrow (2,1)$   via   a:(1, 1, -1, 1)

## C   PROOF THAT MIK+A RECOVERS THE FULL AGENT-CENTRIC STATE

The result in (Lamb et al., 2022) showed that given all examples of the multi-step inverse model from 1 to the diameter of the MDP, achieves the full agent-centric state in a deterministic MDP. Our claim is a reduction to this proof.

## D   ADDITIONAL EXPERIMENTAL DETAILS AND ENVIRONMENT DETAILS

### D.1   POINTMASS ENVIRONMENT DETAILS

#### D.1.1   TOP VIEW

The navigation environments were based on *maze2d-umaze* from D4RL Fu et al. (2020). The state of the navigation is four-dimensional, including the pointmass' position $x, y$ and velocities $v_x, v_y$. The action space is acceleration in each dimension, $a_x, a_y$. For our observation, we disable rendering of the goal in the environment, render images from a camera at the top of the maze, and down-scale them to 100x100. For the curtain experiments, the environment was modified to include both one and three visual occlusions. Environments are visualized in Figure 3.

The data is collected using the built-in planner with Gaussian noise added to the actions. Rather than sampling goals from the fixed set of goals in D4RL, we allow goals to be sampled uniformly at random inside of the maze. The data is collected with no resets, and goals are re-sampled when the pointmass is sufficiently close to the target position.

The original environment is partially observable because it cannot capture the velocity of the point-mass in a single frame. In addition to lacking velocity, the curtain environments also contain regions where the pointmass is partially or fully occluded, and thus the position of the pointmass is not observed. With three curtains, there is additional uncertainty in the position of the pointmass with a single frame, as the pointmass could be under any of the three different curtains. A trajectory can be seen in Figure 10.

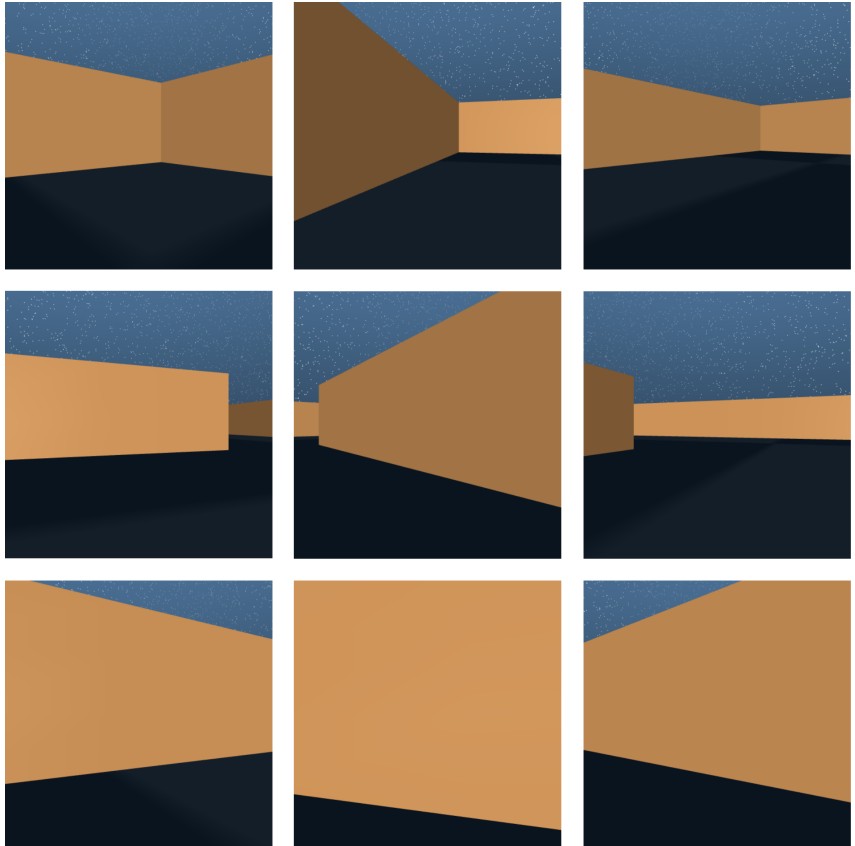

Figure 9: Observations from the first person view environment. Unlike the top view, the global position of the pointmass cannot always be directly inferred from a single observation. Additionally, the maze has many different states with similar looking observations.

### D.1.2 FIRST PERSON VIEW

In addition to the original D4RL environment, we create a new environment that navigates the maze from a first person view (FPV). To do this, we add an angle, $\theta$ and angular velocity, $v_\theta$ to the pointmass' state and change the action space to angular velocity, $v_{\tilde{\theta}}$ and acceleration along the axis the pointmass is facing, $a_x$. The Cartesian velocities $v_x, v_y$, are computed as:

$$v_x \mathrel{+}= a_x \cdot \cos\theta \cdot \Delta t, \; v_y \mathrel{+}= a_x \cdot \sin\theta \cdot \Delta t$$

where $\Delta t$ the angular velocity is set to $v_{\tilde{\theta}}$ and the MuJoCo simulator is stepped for *frame_skip* timesteps. We render images from a camera on the pointmass facing the same direction as the axis of acceleration and use that as our observation.

Since the action space has changed, we train a policy to navigate to goals using PPO Schulman et al. (2017) and use the resulting policy to collect our dataset. Goals and images are modified the same as in the top view environment.

The FPV environment is partially observable for a number of reasons. Like the top view, the velocities cannot be inferred from a single observation. In this environment, however, the global location is not necessarily inferable from any one frame. As can be seen in Figure 9, there are a number of different states in the maze with a similar observation. Having a history of previous observations is required to keep track of the position. Figure 10 shows four frames from the environment.

### D.1.3 EXPERIMENT IMPLEMENTATION DETAILS AND EXTRA EXPERIMENT RESULTS

The total sample size of the offline data for training is 500k for each navigation environment, where each sample is a $(100 \times 100 \times 3)$-dimensional image. At each training iteration, we randomly sample

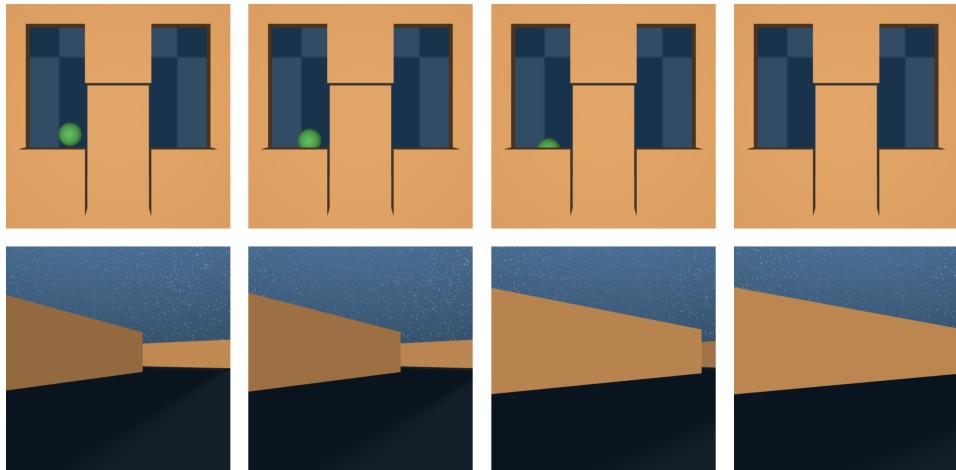

Figure 10: Trajectories from the three curtain and first person environments.

16 batches with time horizon 64 in each batch. The total number of iterations of training is 200k. All the numbers presented in the tables and Figure 4 is the average over the last 10k iterations of the training process. The state estimation errors are the average absolute errors between the true states and the estimated states. The results across different baselines and different navigation environments are provided in Table 3. Since all the numbers is bounded by 1, for better visualization, we provide a barplot of state prediction accuracy (defined as $(1 - \text{state estimate error}) \times 100$) in Figure 4. With decomposing the state error into the position (observable) and velocity (observable) errors, the results are provided in Table 4.

| Objective | $K_{max}$ | No-Curtain | One-Curtain | Three-Curtains | First Person |
|-----------|-----------|------------|-------------|----------------|--------------|
| No History | 1 | 49.3 | 49.9 | 53.7 | 11.8 |
| No History | 15 | 46.8 | 49.1 | 52.6 | 11.4 |
| AH | 1 | 8.6 | 12.7 | 12.8 | 5.3 |
| AH | 15 | 8.9 | 12.0 | 11.8 | 4.6 |
| AH+A | 1 | 44.3 | 45.6 | 46.8 | 25.3 |
| AH+A | 15 | 19.1 | 20.0 | 18.7 | 18.3 |
| FJ | 15 | 7.3 | 12.9 | 13.6 | 5.3 |
| FJ+A | 15 | 3.7 | 5.4 | 6.2 | 5.4 |
| MIK | 1 | 15.3 | 16.4 | 16.2 | 6.0 |
| MIK | 15 | 6.6 | 12.3 | 13.1 | 4.9 |
| MIK+A | 1 | 12.5 | 8.1 | 8.0 | 6.0 |
| MIK+A | 15 | 3.4 | 4.7 | 6.0 | 4.8 |

Table 3: State Estimation Errors (%) on various tasks with maxk=1 vs. maxk=15, with SP=False, no-exo.

In terms of the optimizer, we use Adam optimizer to optimize the losses with learning rate 1e-4. To avoid overfitting, we add $L2$ regularization on forward-running states with the decaying regularization amount. In terms of the neural networks we are training on, there are three neural networks (Encoder, Probe, Action Prediction) with details provided in the following.

**Encoder** The images are encoded using MLP-Mixer (Tolstikhin et al., 2021) with 8 layers and patch size 10, and GRU (Chung et al., 2014) with 2 layers and hidden size 256 is used as the sequence model. For MIK and MIK+A, there is a 2nd GRU network which runs backwards for decoding the future. Alternatively, using a bidirectional RNN for the future works roughly equally well but requires a slightly more involved implementation.

**Probe** We use a 2-hidden layer MLP with hidden size 256 aiming to train a mapping from the latent states to the true states, which in our case is a 4-dimensional vector containing the position

| Objective | P/V | No-Curtain | One-Curtain | Three-Curtains | First Person |
|---|---|---|---|---|---|
| No History | P | 2.2 | 6.5 | 13.3 | 7.7 |
| AH | P | 7.5 | 7.1 | 5.6 | 5.0 |
| AH+A | P | 22.9 | 24.0 | 22.4 | 25.1 |
| FJ | P | 3.2 | 7.1 | 7.1 | 5.0 |
| FJ+A | P | 2.2 | 5.1 | 6.4 | 5.5 |
| MIK | P | 2.5 | 6.1 | 6.5 | 4.7 |
| MIK+A | P | 1.7 | 3.7 | 6.2 | 4.9 |
| No History | V | 91.5 | 91.7 | 91.9 | 15.2 |
| AH | V | 10.0 | 16.9 | 18.1 | 4.3 |
| AH+A | V | 15.2 | 16.0 | 15.0 | 11.5 |
| FJ | V | 11.3 | 18.6 | 20.1 | 5.7 |
| FJ+A | V | 5.2 | 5.8 | 6.0 | 5.4 |
| MIK | V | 10.7 | 18.1 | 19.5 | 5.0 |
| MIK+A | V | 5.0 | 5.6 | 5.7 | 4.7 |

Table 4: Position and Velocity Estimation Errors (%) on various tasks with no exogenous noise, with no self-prediction loss, and with maxk=15.

and velocity of the agent. The loss for the probe is square loss, and no gradients pass from the probe to the representation, such that the use of the probe does not affect the learned representation.

**Action Prediction**   To train a mapping from the current and next latent state to the current action, we use an embedding layer to embed the discrete variable, and then apply a 2-hidden layer MLP with hidden size 256 with the cross entropy loss. In processing the sequence, we compute the action prediction loss for all pairs of time-steps in parallel, thus covering all $k$ values up to a hyperparameter maxk. Experimentally we investigated both maxk=1 and maxk=15. Action prediction losses are provided in Table 5. AH+A contains all the history which is a trivial question for action prediction so that it has almost 0 loss. One optional extra step is to add Self-Prediction (SP) here, by doing this, we use a 3-layer MLP with hidden size 512. The self-prediction objective allows gradients to flow into $s_t$ but blocks gradients into $s_{t+k}$. Adding self-prediction leading to improvement as shown in Table 6. Importantly, it somewhat reduces the degree of difference seen between the various objectives but the overall ordering of results is similar to when self-prediction is not used.

| Objective | No-Curtain | One-Curtain | Three-Curtains | First Person |
|---|---|---|---|---|
| No History | 230.8 | 266.9 | 281.0 | 75.4 |
| AH | 9.2 | 54.9 | 61.7 | 12.4 |
| AH+A | 0.5 | 0.5 | 0.5 | 0.4 |
| FJ | 143.4 | 182.7 | 191.5 | 57.8 |
| FJ+A | 123.7 | 134.6 | 136.9 | 57.9 |
| MIK | 118.5 | 162.3 | 171.4 | 46.4 |
| MIK+A | 99.6 | 112.5 | 114.1 | 37.4 |

Table 5: Action-Prediction Loss (%) with Various Objectives.

**Exogenous Noise**   To construct exogenous noise, we randomly sample images from the CIFAR-10 dataset (Krizhevsky et al., 2014) as the exogenous noise and add them to the original images. Some examples of the no-curtain navigation environment with exogenous noise are provided in Figure 11 as an illustration. The results are provided in Table 7.

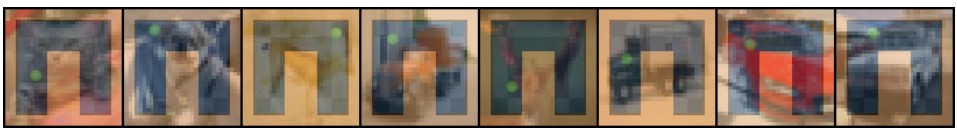

Figure 11: No-curtain navigation environment with exogenous noise.

| Objective | Self-Prediction | No-Curtain | One-Curtain | Three-Curtains | First Person |
|-----------|-----------------|------------|-------------|----------------|--------------|
| No History | N | 46.8 | 49.1 | 52.6 | 11.4 |
| No History | Y | 46.9 | 48.4 | 52.3 | 11.3 |
| AH | N | 8.9 | 12.0 | 11.8 | 4.6 |
| AH | Y | 4.1 | 7.9 | 9.2 | 9.9 |
| AH+A | N | 19.1 | 20.0 | 18.7 | 18.3 |
| AH+A | Y | 18.7 | 17.9 | 19.4 | 11.5 |
| FJ | N | 7.3 | 12.9 | 13.6 | 5.3 |
| FJ | Y | 6.2 | 10.1 | 10.9 | 3.8 |
| FJ+A | N | 3.7 | 5.4 | 6.2 | 5.4 |
| FJ+A | Y | 3.3 | 3.6 | 3.8 | 4.1 |
| MIK | N | 6.6 | 12.3 | 13.1 | 4.9 |
| MIK | Y | 6.0 | 9.9 | 10.7 | 3.7 |
| MIK+A | N | 3.4 | 4.7 | 6.0 | 4.8 |
| MIK+A | Y | 3.3 | 3.5 | 3.8 | 3.8 |

Table 6: State Estimation Errors (%) on various tasks without exogenous noise and with maxk=15, where we show the effect of adding the self-prediction objective, which generally improves the quality of results but leaves the ordering of the methods' performance mostly unchanged.

| Objective | No-Curtain | One-Curtain | Three-Curtains | First Person |
|-----------|------------|-------------|----------------|--------------|
| No History | 47.6 | 49.2 | 52.7 | 12.1 |
| AH | 9.9 | 13.3 | 13.2 | 4.9 |
| AH+A | 18.8 | 18.8 | 18.9 | 18.3 |
| FJ | 10.0 | 14.7 | 15.3 | 5.7 |
| FJ+A | 5.8 | 7.1 | 7.2 | 5.8 |
| MIK | 10.1 | 14.4 | 14.7 | 5.3 |
| MIK+A | 6.1 | 7.1 | 7.4 | 5.4 |

Table 7: State Estimation Errors (%) on various tasks with exogenous noise, and with no-SP, with maxk=15.

## D.2 Offline RL Experiment Details and Setup

We include details on our offline RL experiments including partial observability in the datasets. Figure 12 shows few sample observations from the Cheeta-Run domain when adding random patches to each observation. For our experiments, we use the visual dataset v-d4rl Lu et al. (2022) and to make it partially observable, we randomly add patches of size $(16 \times 16)$ to each observation. This makes the observations non-Markovian in general, such that it is difficult to learn the agent-centric state directly from the observations.

For the experimental setup, we follow the same pre-training of representations procedure from Islam et al. (2023), where we train the encoders learning latent space during pre-training. We then follow the fine-tuning procedure using the fixed representations from the encoders (keep them frozen) and then do offline RL (specifically TD3 + BC) on top of the learnt representations. We use TD3 + BC since it has been already shown to be a minimalistically useful algorithm to learn from offline datasets. We do not use any other offline RL algorithm since in this work, we mainly prioritize on the ability of the encoders to be able to recover the agent-centric state.

Our results show that the inverse kinematics based objectives can be quite useful for recovering the agent-centric state, as stated and justified from our theoretical results; and experimental results show that by using a forward-backward sequence model to handle past and future observations, such inverse kinematics based objectives can be useful especially in presence of non-Markovian observation spaces. Our expeirmental results are indeed quite better compared to the recently proposed ACRO method Islam et al. (2023) on such visual offline datasets. We study the ability of the learnt encoders to be able to learn robust representations from partially observable offline datasets.

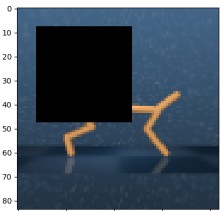 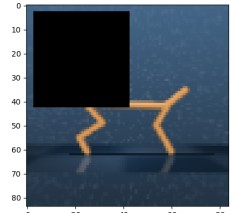 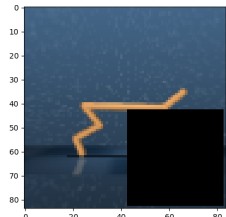

Figure 12: Illustration of patched observations from the visual offline datasets, adapted from Lu et al. (2022). In addition, during frame-stacking when learning from pixel-based observations, we also randomly add zero padding to 2 out of 3 of the stacked frames, to make the pixel-based offline RL setting even more challenging.

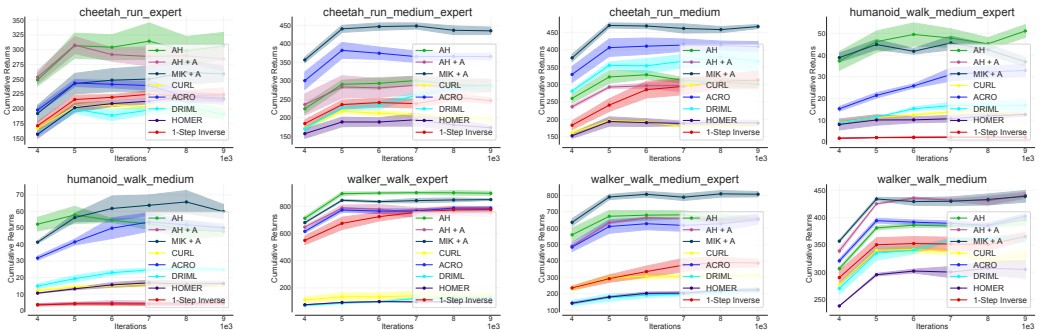

Figure 13: **Patching of Observations :** Full of Experimental Results comparing All the Inverse Kinematics based objectives to other baselines on the $16 \times 16$ patched observation setup.

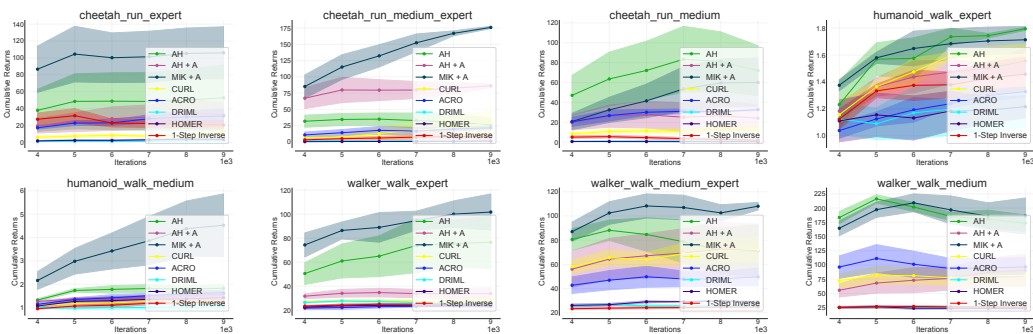

Figure 14: **Random-Zeroing of FrameStacks :** Full of Experimental Results comparing All the Inverse Kinematics based objectives to other baselines on the randomly zeroing of framestacks setup, where in addition to this, we also apply patching of size $16 \times 16$.

# E  COMPARISON WITH DREAMER, DEEPMDP, AND AIS

We further compare the performance of our model with Dreamer Hafner et al. (2019a), AIS Subramanian et al. (2022) and DeepMDP Gelada et al. (2019)

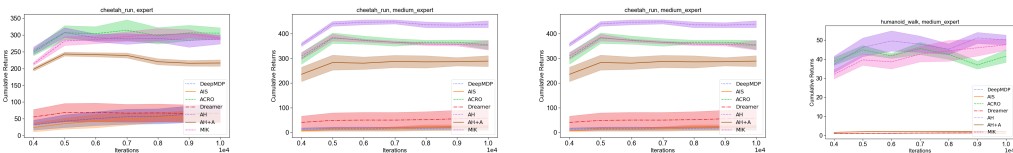

