# OpenReview forum: "Agent-Centric State Discovery for Finite-Memory POMDPs"
_ICLR.cc/2024/Conference — Submitted to ICLR 2024_

### Official Review · Reviewer_HgfR · 2023-10-22

**Soundness:** 3 good
**Presentation:** 2 fair
**Contribution:** 3 good
**Rating:** 5
**Confidence:** 3

**Summary:**

This paper studies agent-centric state discovery in finite-memory partially observable Markov decision processes (POMDPs). The authors consider a setting where the observation space is high-dimensional and non-Markov, but the unobserved latent can be decoded from a sequence of past observations. The state transition dynamics are assumed to factorize into two parts, a deterministic agent-centric part and a control-endogenous part. The authors presented positive results where generalized inverse models can be used for learning agent-centric state representation with some asymptotic analysis, as well as negative results where alternative intuitive algorithms can fail. Experimental results are conducted on a navigation task and a partially observable variant of the vd4rl benchmark.

**Strengths:**

This paper studies the interesting topic of learning an informative state in POMDPs. The authors considered a diverse set of possible inverse kinematic based objectives and reasoned about their pros and cons. The experiment results also look encouraging and validate the authors’ arguments. The objective of “learning state representations” perfectly fits into the scope of ICLR.

**Weaknesses:**

My biggest concern is about the problem setup and especially the assumptions made in the paper. The assumption that the state transition dynamics can factorize into a deterministic agent-centric part and a control-endogenous part seems not very practical to me and needs more motivation. The assumption of a finite agent-centric state space is also not very practical.

In addition, the authors posed the main objective of the paper as “discovering the informative Markovian state space”, yet Section 3 mainly talks about predicting an *action* from a sequence of observations. I am not able to make the connection between these two, especially when the behavior policy is not assumed to be known.

For the theory part, it seems to me that the authors’ analysis heavily relies on a reduction to an existing work (Lamb et al., 2022). It is hence unclear to me what the new contributions are in this work.

The experiment setup seems a bit simplified. In Section 4.1, the authors considered a simple navigation task that is 1-step past-decodable. The authors also only provided comparisons with their own alternative methods but no comparison with other existing works is given. In Section 4.1, the authors modified the vd4rl tasks into POMDPs by randomly masking the observations, which is also a bit artificial to me. I would suggest considering more standard POMDP benchmarks.

The writing and presentation of the paper are not always clear and can sometimes be confusing.

**Questions:**

1.	Could you provide more motivation for the assumptions that I mentioned in the Weaknesses section?

2.	How is “predicting an action from a sequence of observations” related to “discovering the informative Markovian state space”?

3.	Several existing works also consider learning an “approximate information state” in POMDPs, which I believe are relevant and could be discussed in the related work section:
(a)	Subramanian, J., & Mahajan, A. Approximate information state for partially observed systems, 2019.
(b)	Mao, W., Zhang, K., Miehling, E., & Başar, T. Information state embedding in partially observable cooperative multi-agent reinforcement learning, 2020.

4.	What is the $q(· | z)$ function at the end of Page 2. Do you intend to mean the emissions (which has been denoted as $\mathbb{O}$)?

5.	In Page 3, the definition that $\tilde{o}_F(h,n) = \tilde{o}_{\min \{1,h+n\}, H}$ does not look correct.

6.	In Section 4.1, what are the exact metrics that you use to define the “state estimation errors”?

7.	Since the authors only considered the offline RL setting (in the experiments), I would suggest adding the word “offline” to the title or abstract to make the scope of the paper clear to the readers.

8.	Since the setup and methodology of the paper are closely related to several existing works, I would suggest moving the “Related Work” section to an earlier position in the paper.

---

> ### Author Response · Authors · 2023-11-23
> **Response**
>
> > My biggest concern is about the problem setup and especially the assumptions made in the paper. The assumption that the state transition dynamics can factorize into a deterministic agent-centric part and a control-endogenous part seems not very practical to me and needs more motivation.
>
> I think it's not a very strong assumption, because we aren't assuming something like an axis-wise decomposition into agent-centric and exogenous parts.  Rather, we assume that there exists some (unknown) encoders for the agent-centric part and the exogenous parts.  These can be very complicated and non-linear, to allow this factorization to hold.
>
> >The assumption of a finite agent-centric state space is also not very practical.
>
> The finite state assumption is important for theory because it clarified what it means for the state to be "minimal", which is more complicated in continuous spaces.  But for practical purposes requiring the state to be finite is not important.
>
> > In addition, the authors posed the main objective of the paper as “discovering the informative Markovian state space”, yet Section 3 mainly talks about predicting an action from a sequence of observations. I am not able to make the connection between these two, especially when the behavior policy is not assumed to be known.
>
> Predicting the actions is what provides the signal for learning the representations, which we prove constitute a Markovian state space if the objective is satisfied.
>
> > For the theory part, it seems to me that the authors’ analysis heavily relies on a reduction to an existing work (Lamb et al., 2022). It is hence unclear to me what the new contributions are in this work.
>
> The important thing is that we extend multi-step inverse models to POMDPs in a principled way, and we show that several naive generalizations fail to work in theory for POMDPs.
>
> > The authors also only provided comparisons with their own alternative methods but no comparison with other existing works is given.
>
> We ran new experiments with approximate information state (AIS) and Dreamer, and found that both perform relatively poorly on problems with heavy exogenous noise.  Additionally, AIS is very reward-based while the multi-step inverse model does not require rewards.  We will add these results to the paper.  We also have CURL, DRIML, and Homer as contrastive baselines in the paper.

---

### Official Review · Reviewer_Yoch · 2023-10-31

**Soundness:** 4 excellent
**Presentation:** 3 good
**Contribution:** 3 good
**Rating:** 8
**Confidence:** 3

**Summary:**

Building upon a Finite-Memory Partially Observable Markov Decision Process, the authors compare the capabilities of various inverse kinematic approaches for recovering of the agent-centric state in high-dimensional non-Markovian scenarios. Revealing recovery insufficiencies they formally show the applicability of a Masked-Inverse Kinematics objective with the addition of actions. Using this objective, they empirically demonstrate the capability of reconstructing obfuscated states given a sufficient horizon and learning from partially observable image-based offline RL datasets.

**Strengths:**

Overall, the paper is well motivated and articulated. All deductions are well comprehensible and an effective solution approach for a relevant issue is derived with formal justification as well as extensive empirical evaluation.

**Weaknesses:**

Event though, the paper comprehensively covers a relevant topic, some parts could be improved. Most importantly, known limitations should be stated. Also, whilst formally well-motivated, I am missing a notion of the integration of the considered objectives into the experimental setup shown visually in Fig. 5, especially regarding the "SP", "No SP" settings. Also, relations between the masked gap $k$ and the maximum prediction span $K$ could be discussed. Furthermore, the following minor issues in Section 2 should be addressed:

- For convenience $\mathcal{Z}$ should be intoduced as state space and be part of $\mathcal{M}$ (p. 2).
- If not mistaken, $\mathbb{P}(z'|z,a)$, should include $h$ as previously denoted (p. 2).
- Later, z is defined as $z=(s,\xi)$. Thus, $\mathbb{O}$ could also be formalized using $z$ for convenience (p. 2).
- An explanation of $q(o|z)$ is missing (p. 2).
- T is not defined (p. 3).
- $D$ is not introduced (p.  3).
- From my understanding, $\tilde{o}_{F}$ should be bound by h, not 1 (p. 3).
- $\tilde{i}$ should be $\tilde{o}$ (p. 3).
- Table 1: MIK Objective should depend on $o$ not $\tilde{o}$ (p. 6).

Regarding further areas of related research, the authors could consider referring to the artificial obfuscation of fully observable state spaces to partial observability for improved generalization, e.g.:

- M. Laskin, K. Lee, A. Stooke, L. Pinto, P. Abbeel, and A. Srinivas, ‘Reinforcement Learning with Augmented Data’, in Advances in Neural Information Processing Systems, 2020, vol. 33, pp. 19884–19895.
- R. Raileanu, M. Goldstein, D. Yarats, I. Kostrikov, and R. Fergus, ‘Automatic Data Augmentation for Generalization in Reinforcement Learning’, in Advances in Neural Information Processing Systems, 2021, vol. 34, pp. 5402–5415.
- P. Altmann, F. Ritz, L. Feuchtinger, J. Nüßlein, C. Linnhoff-Popien, and T. Phan, ‘CROP: Towards Distributional-Shift Robust Reinforcement Learning Using Compact Reshaped Observation Processing’, in Proceedings of the Thirty-Second International Joint Conference on Artificial Intelligence, IJCAI-23, 8 2023, pp. 3414–3422.

Finally, to foster reproducibility, the authors should consider open-sourcing their implementations upon publication.

**Questions:**

As not clearly stated in the motivation, could you clarify the scope of applicable state spaces?

What are the implications of assuming past- and future-decodability? How do they limit the applicability of the proposed approach?

How does the addition of the two dynamic models impact the computational expense when applying the proposed approach?

Minor Comments:

- Providing examples for non-Markovian environments in the introduction could be helpful.
- p.1: "for an FM-POMDP" -> "for a FM-POMDP"
- Figures and Tables could placed closer to their reference (e.g., Table 1).
- p.5: "in previous" -> "in the previous"
- p.5: Even though refering to Lamb et al. (2022), Theorem 5.1 should be provided.
- Figures 6 and 7: To improve the readability, font sizes should be increased. Shaded areas should be described.

---

> ### Author Response · Authors · 2023-11-23
> **Response**
>
> Thank you for your review.  We appreciate that you took the time to review the paper and provide feedback.
>
> > Also, whilst formally well-motivated, I am missing a notion of the integration of the considered objectives into the experimental setup shown visually in Fig. 5, especially regarding the "SP", "No SP" settings.
>
> We simply add the self-prediction objective to the usual objective in the "SP" case, with a weighting on the self-prediction objective.  We will add these experimental details to the appendix.
>
> > Furthermore, the following minor issues in Section 2 should be addressed:
>
> Thanks for pointing out these writing issues - they will be fixed in the camera ready.
>
> > As not clearly stated in the motivation, could you clarify the scope of applicable state spaces?
>
> For the purposes of theory, we assume that the state space is discrete, but empirically this is definitely not necessary.  Another important requirement is that every state in the MDP should be reachable from every other state in the MDP with a finite number of steps.
>
> > What are the implications of assuming past- and future-decodability? How do they limit the applicability of the proposed approach?
>
> Past-decodability is a reasonable assumption, especially if we make the history very long.  However it rules out some hard-information games such as Poker or Starcraft, where one needs to estimate the state imperfectly based on limited information.  The algorithm could still do something reasonable here but it's beyond the theory.  Future-decodability has some fairly compelling exceptions, which include problems where there is some destruction of information.  For example if I'm playing a video game and then I turn the game off without saving, then future-decodability is violated.
>
> > How does the addition of the two dynamic models impact the computational expense when applying the proposed approach?
>
> By far the largest computational cost is encoding the images into a representation vector, so in the problems that we consider, the computational cost of the dynamics model is negligible.

---

> ### Comment · Reviewer_Yoch · 2023-11-23
>
> Thank you for the clarifications and pointing out intended revisions. Yet, the addressed limitations regarding the applicable state spaces and the assumptions of past- and future-decodability (as also raised by other reviewers) should also be clearly stated in a final version.

---

### Official Review · Reviewer_z5xp · 2023-10-31

**Soundness:** 2 fair
**Presentation:** 2 fair
**Contribution:** 1 poor
**Rating:** 1
**Confidence:** 4

**Summary:**

The paper describes a method for representation learning in POMDPs. It makes two assumptions to ensure states can be recovered from finite observation sequences (both past and future). It then considers several possible approaches for using inverse dynamics models to predict a particular action given a (possibly masked) trajectory of observations (and possibly actions) surrounding that action. The paper claims that the proposed method, which requires looking at specific past and future observations, is the only method among those considered that removes all exogenous variables from the state while retaining all information useful for control. They then present evidence that their method produces a state representation that leads to improved offline RL performance.

**Strengths:**

The paper addresses an important and useful problem:
> *"Can we devise an unsupervised learning algorithm for automatically discovering the informative Markovian state space from a rich non-Markovian observation space?"*

A complete answer to this question would allow agents to employ learning methods that solve MDPs rather than POMDPs, and hence would make decision-making vastly more tractable.

Moreover, the paper's focus on specifically recovering only what the authors call the "agent-centric" state (i.e. only the problem features relevant for decision making) and discarding the irrelevant information, is a longstanding goal of representation learning. A method for reliably accomplishing this would be extremely useful.

**Weaknesses:**

### Main objection

The paper unfortunately suffers from a major weakness. In addition to the standard k-order Markov assumption on past trajectories, it also makes the surprising and confusing choice to require that the hidden state can also be fully decoded from *future* sequences of observations. While this might have interesting mathematical consequences, it also means that the resulting algorithm, which leverages this assumption to decode state based on future observations, cannot lead to a practical decision-making agent.

It is of course true that by using future observations, the agent can more accurately recover the true underlying state of the POMDP. But this fact is completely irrelevant for decision making.

Consider the classic Tiger problem introduced by Kaelbling, Littman, and Cassandra (1998), where the agent must listen for some number of time steps in order to reliably ascertain which of two doors contains a tiger so that it can select the other door that is safe to enter. The original version of the problem returns the agent to the starting state immediately after choosing a door, but suppose we extend the problem by an additional time step so that the resulting state after selecting a door would produce different observations depending on the presence or absence of the tiger. This would satisfy the assumptions in the paper under review.

This paper effectively proposes that the agent simply wait until it has observed whether or not the room it will eventually enter contains the tiger in order to decide what state it is in. But this completely misses the fact that by the time the agent has observed the subsequent room, the crucial decision of selecting which door to open will have already passed!

It is not at all surprising that the authors are able to replace observations with these decoded states and measure improved offline RL performance, because they are providing the agent with more information than it originally had access to. The issue is that this approach can in principle never be useful for online decision making.

### Other Concerns
1. The paper is trying to simultaneously tackle the problems of learning a Markov state abstraction and disentangling the agent-relevant features from the non-relevant ones. This adds complexity, and I don't see much reason to tackle these problems in the same paper.
2. >"the one-step inverse objective learns an undercomplete representation. Intuitively, it may incorrectly merge states which have locally similar dynamics but are actually far apart in the environment."

    I'm not sure this is the correct interpretation. See Allen et al.'s (2021) "Learning Markov State Abstractions for Deep RL", which explores the same idea for MDPs and discusses inverse dynamics models in depth. In particular, they show that even for MDPs, inverse models alone are insufficient for learning an abstract Markov representation. It seems that the approach presented in the submitted paper only works because of the inclusion of future information in the decoding process.

    Furthermore, the suggestion that an inverse model can recover all the agent-state features seems incorrect---it may recover all *controllable* features, but what about non-controllable objects that nevertheless affect either the dynamics or the reward? See Thomas et al.'s (2017) "Independently Controllable Factors" and Sawada et al.'s (2018) follow-up "Disentangling Controllable and Uncontrollable Factors".
3. The paper never actually defines inverse models anywhere, and the "algorithm" (mentioned in the section Forward-Jump Inverse Kinematics (FJ) objective) is only hinted at, never actually presented, even in the appendix.
4. > *"... learning in the presence of both non-Markovian and high-dimensional data. To the best of our knowledge, no prior work has focused on developing techniques for state space discovery in such a setting."*

    I don't understand what the authors could mean by this. It seems to me that there are many such examples. See Ha & Schmidhuber's "World Models", Hafner and colleagues' series of PlaNet and Dreamer models, Guo et al's (2020) Predictions of Bootstrapped Latents (PBL), Subramanian et al's (2022) Approximate Information State (AIS), Zhang et al.'s (2019) causal states.

5. I don't find the experimental results compelling either. Why mask out random frames in the frame stack rather than simply truncating the framestack to use the most recent frame? Why use offline RL? The paper claims that the proposed method "can better capture sufficient information from the partial observations to fully recover the agent-centric state." Was there an experiment that actually measured this? And how is it fair to compare cumulative return relative to methods that can't look into the future?

**Questions:**

So far I cannot see the value of this approach, since it requires the agent to look into the future to construct its state representation, therefore it cannot make decisions based on that representation. Is there some application that I'm missing here? Why would this type of decoding be a valuable thing to do?

---

> ### Author Response · Authors · 2023-11-23
> **Response Part 1**
>
> Thank you for taking the time to review.
>
> >The paper unfortunately suffers from a major weakness. In addition to the standard k-order Markov assumption on past trajectories, it also makes the surprising and confusing choice to require that the hidden state can also be fully decoded from future sequences of observations. While this might have interesting mathematical consequences, it also means that the resulting algorithm, which leverages this assumption to decode state based on future observations, cannot lead to a practical decision-making agent
>
> I believe there is a severe misunderstanding here.  In our method, we make a past-decodability assumption and a future-decodability assumption, and these are used as the basis for two encoders which are *BOTH* capable of encoding the state: one encodes using the past and one encodes using the future.  During training, you sample from the replay buffer and can use both encoders for training.  During inference (planning/exploration/etc) only the past-encoder is used.
>
> Experimentally when we measure whether the encoder has information about the state, we are evaluating the past-encoder only.
>
> We will revise the draft for the camera ready to make this point more forcefully, because it’s critical for understanding the method correctly.
>
> > Consider the classic Tiger problem introduced by Kaelbling, Littman, and Cassandra (1998), where the agent must listen for some number of time steps in order to reliably ascertain which of two doors contains a tiger so that it can select the other door that is safe to enter.
> > This paper effectively proposes that the agent simply wait until it has observed whether or not the room it will eventually enter contains the tiger in order to decide what state it is in. But this completely misses the fact that by the time the agent has observed the subsequent room, the crucial decision of selecting which door to open will have already passed!”
>
> Once you have heard the tiger roar, you can satisfy past decodability (the state can be decoded from the past) and once the observation shows the position of the tiger, you can satisfy future decodability as well.  Note that past-decodability can be achieved once the roar is heard, so the past-encoder can be used to plan intelligently.  Future-decodability is only achieved when the door is opened, but the future-decodability assumption (for the future-encoder) is only used during training, as mentioned in the answer to the previous question.
>
> It’s not clear to me that this is a significant issue.  If the episode really resets immediately after opening the door, then in the reward-free setting (which is what we concern ourselves with) then it doesn’t matter for the problem which door is opened.  On the other hand, if we have reward and we simply append it to our observation once it’s received, then we recover the workaround that you mentioned of modifying the observation.
>
> > The paper is trying to simultaneously tackle the problems of learning a Markov state abstraction and disentangling the agent-relevant features from the non-relevant ones. This adds complexity, and I don't see much reason to tackle these problems in the same paper.
>
> Both of these problems are fairly well-studied on their own.  The multi-step inverse kinematics approach has been widely explored for learning agent-relevant features and this is the first paper to study how to correctly generalize this objective to the finite-memory partially observable setting.  Intriguingly, we show that several naive generalizations of the multi-step inverse model are ineffective and provide a clear explanation of why they don’t necessarily work correctly.

---

> > ### Comment · Reviewer_z5xp · 2023-11-23
> >
> > >  In our method, we make a past-decodability assumption and a future-decodability assumption, and these are used as the basis for two encoders which are BOTH capable of encoding the state: one encodes using the past and one encodes using the future. During training, you sample from the replay buffer and can use both encoders for training. During inference (planning/exploration/etc) only the past-encoder is used.
> > >
> > > $ $
> > >
> > > Experimentally when we measure whether the encoder has information about the state, we are evaluating the past-encoder only.
> > >
> > > $ $
> > >
> > > We will revise the draft for the camera ready to make this point more forcefully, because it’s critical for understanding the method correctly.
> >
> > $ $
> >
> > This is quite surprising to hear. I specifically remember looking for this during my initial review, and I looked again just now, to try to find any place in the text where it explains that the future information is only used during training. I still cannot find any mention of it.
> >
> > If indeed it is the case that the method is intended to have two phases, and that the agent is allowed to look into the future during training, but not during test, it is critical that this be made clear at the outset.
> >
> > In any case, this revelation makes me more confused. If information from the future is being used to train the representation, I can see how that would lead to better representations. However, that information cannot be useful at inference time, since it isn't available to the inference encoder. How does this method reliably decode state information that isn't available until later in the episode? And if the answer is along the lines that the forward and backward decodability assumptions ensure that it *is* available, then I take issue with those assumptions, because that means they are unrealistic.
> >
> > Again, the tiger problem reveals this asymmetry. My point with the tiger example is that the information from the future would be really nice for the agent to have, but unfortunately all it has are the noisy observations from its past. Recall that listening in the tiger problem only reveals the tiger's correct location ~85% of the time. Thus, consecutive observations are necessary to reduce the uncertainty over which door contains the tiger. In principle, given enough observations, this uncertainty can be reduced to an infinitely small value, but it can never go away completely. By contrast, *visiting* one of the rooms reduces that uncertainty to exactly zero, and incorporating that future information would result in the belief state collapsing to a single door.
> >
> > So even if we train an encoder using information from past *and* future, how can we expect the inference encoder to reliably predict that collapsed belief state, given only information from the past? The paper contains no explanation of how this can work.

---

> ### Author Response · Authors · 2023-11-23
> **Response Part 2**
>
> > See Allen et al.'s (2021) "Learning Markov State Abstractions for Deep RL", which explores the same idea for MDPs and discusses inverse dynamics models in depth. In particular, they show that even for MDPs, inverse models alone are insufficient for learning an abstract Markov representation. It seems that the approach presented in the submitted paper only works because of the inclusion of future information in the decoding process.
>
> The Allen 2021 paper is about combining one-step inverse models with temporal contrastive objectives.  This will capture exogenous noise and thus fail to have the same guarantees as multi-step inverse models.  Moreover, the paper is not about the finite-memory partially observable setting.
>
> > Furthermore, the suggestion that an inverse model can recover all the agent-state features seems incorrect---it may recover all controllable features, but what about non-controllable objects that nevertheless affect either the dynamics or the reward? See Thomas et al.'s (2017) "Independently Controllable Factors" and Sawada et al.'s (2018) follow-up "Disentangling Controllable and Uncontrollable Factors".
>
> Understanding exactly what the multi-step inverse model captures is still an ongoing area of research, and whether it really consists of all controllable features or the complete “agent-state” is a subtle question.  You are correct that it may drop information which is related to the reward.
>
> > I don't understand what the authors could mean by this. It seems to me that there are many such examples. See Ha & Schmidhuber's "World Models", Hafner and colleagues' series of PlaNet and Dreamer models, Guo et al's (2020) Predictions of Bootstrapped Latents (PBL), Subramanian et al's (2022) Approximate Information State (AIS), Zhang et al.'s (2019) causal states.
>
> You are correct, we meant to say “agent-centric state space”.  This will be fixed for the camera ready.
>
> >Why mask out random frames in the frame stack rather than simply truncating the framestack to use the most recent frame?
>
> The masking of random frames is what introduces partial observability in the offline-RL setting we considered.

---

> > ### Comment · Reviewer_z5xp · 2023-11-23
> >
> > I only brought up the Allen et al. paper in the specific context of the one-step inverse model, which was mentioned in your manuscript and which I quoted in my review. I'm saying they specifically studied that kind of 1-step inverse model, and showed (as noted here) that it learns an undercomplete representation. I'm not arguing that point. Rather, I was saying I wasn't sure I agreed with your *interpretation* of why that occurs. Their paper showed that with a 1-step inverse model, an undercomplete representation can occur that merges *adjacent* states together, even when they have *different* dynamics. It's almost the opposite of the interpretation in the text I quoted in my review, which I'll repeat here, for ease of reading:
> >
> > > "the one-step inverse objective learns an undercomplete representation. Intuitively, it may incorrectly merge states which have locally similar dynamics but are actually far apart in the environment."
> >
> > So if this point is true, then at best, it's only part of the story with respect to 1-step inverse models.
> >
> > $ $
> >
> > -----
> >
> > $ $
> >
> > > > Why mask out random frames in the frame stack rather than simply truncating the framestack to use the most recent frame?
> > >
> > > The masking of random frames is what introduces partial observability in the offline-RL setting we considered.
> >
> > This doesn't answer my question. It seems needlessly complicated in comparison to simple truncation. I'm wondering why you chose this type of partial observability rather than a simpler one.

---

### Official Review · Reviewer_38V6 · 2023-11-03

**Soundness:** 2 fair
**Presentation:** 1 poor
**Contribution:** 1 poor
**Rating:** 1
**Confidence:** 3

**Summary:**

The authors propose an unsupervised method for estimating a Markov state from a fixed-length sequence of observations. They use a classifier to select an action from a series of observations, proposing four different objectives. They ablate their method on a navigation task with curtains and and VD4RL tasks with some pixels obscured.

**Strengths:**

- The authors ablate their approach

**Weaknesses:**

- The paper is difficult to read
- The paper claims that unsupervised learning for Markov state discovery is novel -- this is not true, as evidenced by world-model based approaches for POMDPs such as Dreamer
- I'm not familiar with the "agent-centric state" definition central to the paper, and a cursory search returns few results. It seems to be equivalent to the Markov state "which captures all information that is relevant to the control of the agent".
- The literature review seems to be missing lots work on unsupervised objectives in RL

**Questions:**

- Typo in the abstract (double spaces)
- Abstract is not very clear -- what is an agent-centric state? Alternative intuitive algorithms? Etc.
- "Can we devise an unsupervised learning algorithm for automatically discovering the informative Markovian state space from a rich non-Markovian observation space?" - Isn't this model-based RL or a dreamer-style approach?
- "To the best of our knowledge, no prior work has focused on developing techniques for state space discovery in such a setting." - There is prior work, please see [1] or any work on partially observable world models
- "Yet, none explicitly focused on state discovery." - What about [2]?

[1] https://arxiv.org/pdf/1912.01603.pdf

[2] https://proceedings.neurips.cc/paper/2020/hash/3c09bb10e2189124fdd8f467cc8b55a7-Abstract.html

---

> ### Author Response · Authors · 2023-11-23
> **Response**
>
> > The paper claims that unsupervised learning for Markov state discovery is novel -- this is not true, as evidenced by world-model based approaches for POMDPs such as Dreamer
>
> This is an important issue.  In this work we are only focused on multi-step inverse models for discovering agent-centric state while discarding exogenous noise.  Methods like Dreamer predict in the observation space, and thus have no ability to ignore exogenous noise in their representations.  However, there is a Denoised-MDP paper which we cited, which learns both agent-centric state and exogenous noise and then learns to separate them out.
>
> > I'm not familiar with the "agent-centric state" definition central to the paper, and a cursory search returns few results. It seems to be equivalent to the Markov state "which captures all information that is relevant to the control of the agent".
>
> It is discussed in the introduction and related work, but here are a few recent and relevant papers dealing with agent-centric state (sometimes referred to as controllable or endogenous state) and filtering exogenous noise:
>
> https://arxiv.org/pdf/2207.08229.pdf
>
> https://arxiv.org/pdf/2211.00164.pdf
>
> https://openreview.net/forum?id=RQLLzMCefQu
>
> > "To the best of our knowledge, no prior work has focused on developing techniques for state space discovery in such a setting." - There is prior work, please see [1] or any work on partially observable world models
> > "Yet, none explicitly focused on state discovery." - What about [2]?
>
> We ran Dreamer on our Mujoco offline-RL experiments (with very strong exogenous distractors) and indeed found that it performed poorly.  We will update the paper to add these new results.  Temporal distance objectives [2] are also vulnerable to exogenous noise, and we have already included two such baselines in the paper: DRIML and Homer.

---

### Meta-Review · Area_Chair_UQ1g · 2023-12-13

**Metareview:**

This paper considers finite-memory partially observable Markov decision processes (POMDPs) and studies the problem of agent-centric state discovery. The authors focus on a setting where the observation space is high-dimensional and non-Markov, but the unobserved latent state can be decoded from a sequence of past/future observations. The state transition dynamics are assumed to factorize into two parts, a deterministic agent-centric part and a control-endogenous part. The authors presented positive results where generalized inverse models can be used for learning agent-centric state representation with some asymptotic analysis, as well as negative results where alternative intuitive algorithms can fail. To validate these concepts, experimental evaluations are conducted on both a navigation task and a partially observable variant of the vd4rl benchmark. Unfortunately, for the current state of the paper, there remain several major concerns regarding the assumptions being too strong and the novelty/contribution is not enough. Those issues have not been addressed after rebuttal. Therefore, we recommend rejection.

**Justification For Why Not Higher Score:**

Assumption is strong, and not well explained in the paper. Contribution comparing to prior work is not enough.

**Justification For Why Not Lower Score:**

N/A

---

### Decision · Program_Chairs · 2024-01-16

Reject